

# Stress Tensor flows, birefringence in non-linear electrodynamics and supersymmetry

**Christian Ferko[1⋆], Liam Smith[2†] and Gabriele Tartaglino-Mazzucchelli[2‡]**

**1** Center for Quantum Mathematics and Physics (QMAP),
Department of Physics & Astronomy, University of California, Davis, CA 95616, USA
**2** School of Mathematics and Physics, University of Queensland,
St Lucia, Brisbane, Queensland 4072, Australia

⋆ caferko@ucdavis.edu , † liam.smith1@uq.net.au , ‡ g.tartaglino-mazzucchelli@uq.edu.au

## Abstract

We identify the unique stress tensor deformation which preserves zero-birefringence conditions in non-linear electrodynamics, which is a $4d$ version of the $T\overline{T}$ operator. We study the flows driven by this operator in the three Lagrangian theories without birefringence – Born-Infeld, Plebanski, and reverse Born-Infeld – all of which admit ModMax-like generalizations using a root-$T\overline{T}$-like flow that we analyse in our paper. We demonstrate one way of making this root-$T\overline{T}$-like flow manifestly supersymmetric by writing the deforming operator in $\mathcal{N} = 1$ superspace and exhibit two examples of superspace flows. We present scalar analogues in $d = 2$ with similar properties as these theories of electrodynamics in $d = 4$. Surprisingly, the Plebanski-type theories are fixed points of the classical $T\overline{T}$-like flows, while the Born-Infeld-type examples satisfy new flow equations driven by *relevant* operators constructed from the stress tensor. Finally, we prove that any theory obtained from a classical stress-tensor-squared deformation of a conformal field theory gives rise to a related "subtracted" theory for which the stress-tensor-squared operator is a constant.

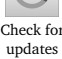

# 1  Introduction

In the past several years, many interesting connections have emerged between special quantum field theories and deformations involving operators constructed from the energy-momentum tensor $T_{\mu\nu}$. By "special" we mean theories which enjoy some additional property such as integrability, conformal invariance, or supersymmetry, or models which emerge naturally from string theory. The most well-studied example of a stress tensor deformation is the $T\bar{T}$ deformation of two-dimensional (2*d*) quantum field theories [1–3]. This perturbation is constructed using the coincident point limit

$$O_{T\bar{T}}(x) = \lim_{y \to x} \left( T^{\mu\nu}(x)T_{\mu\nu}(y) - T^{\mu}{}_{\mu}(x)T^{\nu}{}_{\nu}(y) \right), \tag{1}$$

which can be shown to define a local operator in any translation-invariant 2*d* field theory.

This $T\bar{T}$ deformation has at least three properties which make it especially interesting:

1. The operator is *universal*, in the sense that it takes the form (1) regardless of the details of the seed theory, and because this deformation is available in any translation-invariant 2*d* QFT.

2. The $T\bar{T}$ deformation is *solvable*, insofar as quantities in the deformed theory can often be computed in terms of corresponding quantities in the undeformed theory. Examples include the finite volume spectrum [2,3], flat space $S$-matrix [4], torus partition function [5–7], and correlation functions [8,9].

3. Deforming by $O_{T\bar{T}}$ preserves many *symmetries* and other desirable properties of the seed theory, such as, for example, integrability [2] and supersymmetry [10–17]. For a $T\bar{T}$-deformed CFT, the deformed theory is even invariant under a certain modified (field-dependent) conformal transformation [18–21].

The property of solvability is especially unusual because the operator $O_{T\bar{T}}$ is irrelevant in the sense of the renormalization group, and a deformation by an irrelevant operator is typically not under analytic control. Because of these and other aspects of $T\bar{T}$, several hundred papers have appeared on the subject in the past few years, which we do not attempt to review in detail here. We instead refer the reader to the lectures [22, 23] and references therein for an introduction to the subject.

It is remarkable that $O_{T\bar{T}}$ is always present in the spectrum of local operators and that many properties of $T\bar{T}$-deformed theories can be probed at the quantum level. However, it is

also illuminating to think of this quadratic combination of stress tensors as a classical object and study the flow equation

$$\frac{\partial \mathcal{L}_\lambda}{\partial \lambda} = T^{\mu\nu} T_{\mu\nu} - \left(T^\mu{}_\mu\right)^2 \qquad (d = 2),$$

(2)

for the Lagrangian defining the field theory. Together with an initial condition $\mathcal{L}_0$ which describes the seed theory, the differential equation (2) defines a one-parameter family of Lagrangians $\mathcal{L}_\lambda$ labeled by the deformation parameter $\lambda$.

Many interesting theories arise from solving the flow equation (2). Perhaps the most striking result, as we will review in Section 2, is that the solution to this differential equation with an initial condition that describes a free massless scalar, $\mathcal{L}_0 = \partial^\mu \phi \partial_\mu \phi$, is the Lagrangian of a static gauge-fixed Nambu-Goto string in three target spacetime dimensions [3]. This is the first hint of a relationship between the $T\overline{T}$ deformation and string theory, which has been further developed in many directions. For instance, the high-energy density of states of a $T\overline{T}$-deformed CFT is Hagedorn and a single-trace version of the $T\overline{T}$ deformation is related to little string theory [24–26], the operator $O_{T\overline{T}}$ can be linked to the uniform light-cone gauge [10,27,28], and there is a proposal for a non-perturbative definition of $T\overline{T}$ in terms of non-critical strings [29]. Solutions to the flow equation (2) with other initial conditions have been studied in [30,31].

If we restrict our attention to classical flow equations for the Lagrangian, and do not demand that the corresponding combinations define local operators at the quantum level, then there are a few ways to generalize and extend the 2$d$ deformation (2). One way is to work in higher spacetime dimensions. The most straightforward generalization of the two-dimensional Lagrangian flow equation is

$$\frac{\partial \mathcal{L}}{\partial \lambda} = a T^{\mu\nu} T_{\mu\nu} + b \left(T^\mu{}_\mu\right)^2 \qquad (d \geq 2),$$

(3)

for some suitable choice of dimensionless constants $a, b$. Just as this classical flow generates theories related to strings in $d = 2$ for $a = 1 = -b$, another appropriate choice of parameters appears to generate theories related to branes in $d = 4$ spacetime dimensions: the solution to one such 4$d$ flow with $a = 1 = -2b$ whose initial condition is the free Maxwell theory, $\mathcal{L}_0 = -\frac{1}{4} F^{\mu\nu} F_{\mu\nu}$, is the Born-Infeld action[1] that describes the gauge theory on a brane's worldvolume [34].

A second way to extend the study of such classical flows is to consider deformations by other combinations of stress tensors. One possibility is to consider a marginal flow

$$\frac{\partial \mathcal{L}}{\partial \gamma} = R,$$

(4a)

$$R = c\sqrt{\hat{T}^{\mu\nu} \hat{T}_{\mu\nu}}, \qquad \hat{T}_{\mu\nu} = T_{\mu\nu} - \frac{1}{d} g_{\mu\nu} T^\rho{}_\rho,$$

(4b)

in any $d \geq 2$, where $\hat{T}_{\mu\nu}$ is the traceless part of the stress tensor and $c$ is another dimensionless constant. In two spacetime dimensions and for $c = \frac{1}{\sqrt{2}}$, this gives the classical root-$T\overline{T}$ flow which has been studied in [35]; see also [36–42] for related work.

It is not known whether the marginal combination of (4b) leads to a well-defined operator at the quantum level, even in two spacetime dimensions. However, even as a classical deformation of the Lagrangian, the flow generated by this operator $R$ has interesting properties. For instance, this deformation appears to preserve integrability in several 2$d$ models, as one can explicitly write down a deformed Lax connection [43].

---

[1] A modified version of this flow equation, with an additional term that is non-analytic in the stress tensor, generates the Born-Infeld theory in three dimensions [32]. This flow can be supersymmetrized and gives the $\mathcal{N} = 1$ supersymmetric BI theory discussed in [33].

Further, deforming the free Maxwell Lagrangian in four spacetime dimensions by $R$ leads to the Modified Maxwell or "ModMax" theory of non-linear electrodynamics which has been introduced in [44–47], and whose Lagrangian can be written as

$$\mathcal{L}_{\text{ModMax}} = -\frac{1}{4}\cosh(\gamma)F^2 + \frac{1}{4}\sinh(\gamma)\sqrt{(F^2)^2 + (F\widetilde{F})^2}\,. \tag{5}$$

Here $\widetilde{F}^{\mu\nu} = \frac{1}{2}\epsilon^{\mu\nu\rho\sigma}F_{\rho\sigma}$ is the Hodge dual of the Abelian field strength $F_{\mu\nu}$. The ModMax theory is special in the sense that it is the unique deformation of the free Maxwell theory in $4d$ which preserves both conformal invariance and electric-magnetic duality invariance. The preservation of conformal symmetry is in accord with the fact that this theory is obtained as a (classically) marginal deformation of the Maxwell Lagrangian, which is itself conformally invariant. In [48, 49], flow equations were presented for both the ModMax Lagrangian (5) and its extension to the Born-Infeld-ModMax theory whose Lagrangian can be written as

$$\mathcal{L}_{\gamma\text{BI}} = \frac{1}{\lambda}\left\{ 1 - \sqrt{1 + \frac{\lambda}{2}\left[\cosh(\gamma)F^2 - \sinh(\gamma)\sqrt{(F^2)^2 + (F\widetilde{F})^2}\right] - \frac{\lambda^2}{16}(F\widetilde{F})^2} \right\}\,. \tag{6}$$

One can view the theory (6) as a doubly-deformed model which arises from flowing the free Maxwell Lagrangian by both an irrelevant $T\overline{T}$-like operator[2] and a marginal root-$T\overline{T}$-like operator (in either order, as the flows can be shown to commute). For a recent review of theories of non-linear electrodynamics, including the ModMax theory and its Born-Infeld-ModMax extension, see [50].

The motivation for the present work is to address several lingering questions about these classical stress tensor flows. One question is: to what extent do stress tensor deformations preserve special features of the seed theories? For instance, in $d = 4$, both the free Maxwell theory and the Born-Infeld theory which arises as its $T\overline{T}$-like deformation exhibit the special property of exhibiting zero-birefringence. We will see that this is not a coincidence, and in fact the classical $T\overline{T}$-like flow generically preserves this property.

Another special feature that a seed theory might possess is supersymmetry. It is already known that, in many examples, $T\overline{T}$ and other $T\overline{T}$-like deformations can be presented in a manifestly supersymmetric form by writing the perturbing operator in superspace [10–13, 15–17]. Most relevant for the context of four-dimensional gauge theories is the observation of [15] that the $4d$ $T\overline{T}$-like flow can be written as a supercurrent-squared deformation in $\mathcal{N} = 1$ superspace, and that the result of deforming a free vector multiplet is the supersymmetric Born-Infeld action. This was extended to supercurrent-squared deformations of the supersymmetric Born-Infeld-ModMax theory in [49].

Given the especially nice interplay between irrelevant $T\overline{T}$-like deformations and supersymmetry, one is led to wonder: can the marginal root-$T\overline{T}$-like operator in $4d$ also be written in such a manifestly supersymmetry-preserving way? We will see that the answer to this question is also affirmative, at least in certain examples involving supersymmetric gauge theories. This is encouraging because the additional control provided by supersymmetry is most powerful when it is made geometric by such a superspace construction.

A third question concerns the degree to which the stress tensor deformations that generate these special theories are unique. Can one find flow equations driven by other combinations of energy-momentum tensors which theories like Born-Infeld and ModMax also satisfy? Indeed, we will find that many of these theories also obey differential equations driven by *relevant*

---

[2]We will use the terms "$T\overline{T}$-like flows" or "$T^2$ flows" for classical flow equations driven by a quadratic combination of stress tensors in any number of spacetime dimensions, reserving the term "$T\overline{T}$ flow" for $d = 2$. Similarly, we say "root-$T^2$ flow" or "root-$T\overline{T}$-like flow" for marginal stress tensor deformations in any dimension, saying "root-$T\overline{T}$ flow" only in $d = 2$. We will also sometimes use the terms "$\lambda$-flow" for a $T^2$ deformation and "$\gamma$-flow" for a root-$T^2$ deformation.

operators constructed from $T_{\mu\nu}$, unlike the irrelevant $T\overline{T}$ or marginal root-$T\overline{T}$. These relevant flows are generated by adding an appropriate constant term to the Lagrangian which causes the classical $T\overline{T}$-like combination of stress tensor bilinears to become a *constant*, independent of fields.

The layout of this paper is as follows. In Section 2, we will develop some general observations about classical $T\overline{T}$-like deformations, focusing on special theories where the $T^2$ operator is a constant. Section 3 applies these results to several four-dimensional gauge theories, explaining the relationship between stress tensor flows and additional properties such as zero-birefringence conditions and electric-magnetic duality invariance; Section 4 then presents analogues of these theories in two spacetime dimensions. In Section 5, we develop a version of the $4d$ root-$T\overline{T}$-like flow with manifest $\mathcal{N} = 1$ supersymmetry and apply it to two examples. Finally, in Section 6 we summarize these results and identify several directions for future investigation.

## 2  Relevant $T\overline{T}$-like flows and $T^2$ fixed points

In this Section, we will consider certain deformations which arise from combining classical $T\overline{T}$-like flows with the addition of a suitable constant term to the Lagrangian. In a theory with dynamical gravity, this constant term can be interpreted as a cosmological constant whose value is correlated with the $T\overline{T}$ flow parameter, as studied in [51–53].[3]

In our case, we will be motivated by examples in two and four spacetime dimensions where a classical $T\overline{T}$-like flow generates a string or brane action along with a constant term in the Lagrangian. For instance, it is well-known [3] that the solution to the two-dimensional $T\overline{T}$ flow equation

$$\frac{\partial \mathcal{L}}{\partial \lambda} = \frac{1}{4}\left( T^{\mu\nu}T_{\mu\nu} - \left(T^{\mu}_{\ \mu}\right)^2 \right), \tag{7}$$

with initial condition $\mathcal{L}_0 = -\partial^\mu \phi \, \partial_\mu \phi$, is

$$\mathcal{L}^{\mathrm{NG}}_\lambda = \frac{1}{\lambda}\left( 1 - \sqrt{1 + 2\lambda\partial^\mu\phi\,\partial_\mu\phi} \right). \tag{8}$$

The Lagrangian (8) represents a static gauge Nambu-Goto string in three target spacetime dimensions, although the conventional way of writing the Nambu-Goto Lagrangian does not include the constant term $\frac{1}{\lambda}$. From the perspective of the classical $T\overline{T}$ flow, this $\lambda$-dependent constant term is needed to ensure that the deformed Lagrangian correctly reduce to the initial condition $\mathcal{L}_0$ at $\lambda = 0$; physically, one can interpret this term as a worldsheet coupling to a constant target-space $B$ field. We will be interested in the corresponding "subtracted" form of the Lagrangian,

$$\widetilde{\mathcal{L}}_\lambda = \mathcal{L}_\lambda - \frac{1}{\lambda}. \tag{9}$$

Throughout this section, we will use a tilde to denote the subtracted form of any Lagrangian, defining

$$\widetilde{\mathcal{L}} = \mathcal{L} - \frac{1}{\lambda}, \tag{10}$$

for any $\mathcal{L}$.

A similar structure appears in deformations of four-dimensional gauge theories. Let $F_{\mu\nu}$ be the field strength associated with an Abelian gauge field $A_\mu$, and define the two Lorentz

---

[3]Said differently, the constant term in the Lagrangian affects the vacuum energy; the vacuum energy of a $T\overline{T}$ deformed CFT, which is proportional to $\frac{1}{\lambda}$, was considered in the recent analysis of [54].

invariants

$$S = -\frac{1}{4} F_{\mu\nu} F^{\mu\nu}, \qquad P = -\frac{1}{4} F_{\mu\nu} \widetilde{F}^{\mu\nu}, \tag{11}$$

where $\widetilde{F}_{\mu\nu} = \frac{1}{2} \epsilon^{\mu\nu\rho\sigma} F_{\rho\sigma}$ is the Hodge dual of $F_{\mu\nu}$. It is also known [34] that the solution to the four-dimensional stress-tensor-squared flow equation,

$$\frac{\partial \mathcal{L}}{\partial \lambda} = \frac{1}{8} \left( T^{\mu\nu} T_{\mu\nu} - \frac{1}{2} \left( T^{\mu}{}_{\mu} \right)^2 \right), \tag{12}$$

with initial condition $\mathcal{L}_0 = S$, is

$$\mathcal{L}_\lambda^{\mathrm{BI}} = \frac{1}{\lambda} \left( 1 - \sqrt{1 - 2\lambda S - \lambda^2 P^2} \right), \tag{13}$$

which is – again, up to the overall scaling and the addition of the $\lambda$-dependent constant term – the Born-Infeld Lagrangian representing the gauge theory on the worldvolume of a $D$-brane with tension

$$T = \frac{1}{\lambda}. \tag{14}$$

Note that the tension $T$ is not to be confused with the symbol $T$ appearing in $O_{T^2}$, which refers to the stress tensor $T_{\mu\nu}$. As in the two-dimensional case, we will be interested in the subtracted Lagrangian

$$\widetilde{\mathcal{L}}_\lambda^{\mathrm{BI}} = \mathcal{L}_\lambda^{\mathrm{BI}} - \frac{1}{\lambda}. \tag{15}$$

Although the procedure of removing a constant term from the Lagrangian appears trivial, we will see that these subtracted theories possess some unusual properties from the perspective of stress tensor flows. For instance, after performing the subtraction and computing the stress tensor $\widetilde{T}^{\mu\nu}$ of the modified theory, the combination which defines our $T^2$ operator in the modified theory is *constant*:

$$\widetilde{O}_{T^2} = \frac{1}{2d} \left( \widetilde{T}^{\mu\nu} \widetilde{T}_{\mu\nu} - \frac{2}{d} \left( \widetilde{T}^{\mu}{}_{\mu} \right)^2 \right) = -\frac{2}{\lambda^2}. \tag{16}$$

We will see that this constant-$T^2$ property allows us to write a new flow equation for the subtracted theories in terms of a *relevant* operator,

$$\frac{\partial \widetilde{\mathcal{L}}}{\partial T} = \frac{1}{d} \frac{\widetilde{T}^{\mu}{}_{\mu}}{\sqrt{\left| 2\widetilde{O}_{T^2} \right|}} = \frac{\widetilde{T}^{\mu}{}_{\mu}}{\sqrt{\left| d\widetilde{T}^{\mu\nu} \widetilde{T}_{\mu\nu} - 2 \left( \widetilde{T}^{\mu}{}_{\mu} \right)^2 \right|}}, \tag{17}$$

where $T = \frac{1}{\lambda}$. This property is a generic feature of the solutions to $T\overline{T}$-like flows with classically conformal seed theories in any spacetime dimension $d$, and is therefore not special to deformations of free scalars in $d = 2$ or the free Maxwell theory in $d = 4$.

## 2.1 Trace flow equation

To study these subtracted flows, we will need a standard fact about classical $T\overline{T}$-like flows which is often referred to as the trace flow equation.

This trace relation has been used many times in the $T\overline{T}$ literature, especially in the context of cutoff AdS$_3$ holography [55]. For instance, the trace relation can be used to identify the dictionary between the $T\overline{T}$ flow parameter $\lambda$ and the bulk Newton constant $G$ as explained in [9]; this correspondence is further refined in [56–58] where again the $T\overline{T}$ trace flow equation plays an important role. See also Section 5.3 of the lecture notes [22] for a review of the trace relation and its applications.

Although this result is elementary, we will review it here for completeness and to fix our conventions. We first work in a slightly more general setting. Consider a seed theory $\mathcal{L}_0$, in $d$ spacetime dimensions, which is classically conformally invariant. In particular, we assume that there is no characteristic length scale $\ell$ associated with $\mathcal{L}_0$, such as the length $\ell = \frac{1}{m}$ which would be associated with the theory of a massive particle with mass $m$. Let $\mathcal{L}_\lambda$ be the one-parameter family of theories which solves a flow equation driven by any operator $f\left(T_{\mu\nu}(\lambda)\right)$ which is a Lorentz scalar constructed from the stress tensor:

$$\frac{\partial \mathcal{L}_\lambda}{\partial \lambda} = f\left(T_{\mu\nu}(\lambda)\right). \tag{18}$$

For instance, the function $f\left(T_{\mu\nu}(\lambda)\right)$ could be the trace of the stress tensor $T^\mu{}_\mu$, or a bilinear combination such as $a T^{\mu\nu} T_{\mu\nu} + b\left(T^\mu{}_\mu\right)^2$ with adimensional constants $a$ and $b$, or more generally any function of the $d$ independent traces $\mathrm{Tr}\left(T^i\right)$ for $i = 1, \ldots, d$:

$$f\left(T_{\mu\nu}(\lambda)\right) = f\left(\mathrm{Tr}(T), \mathrm{Tr}(T^2), \ldots, \mathrm{Tr}(T^d)\right). \tag{19}$$

Here we have written $T_{\mu\nu}(\lambda)$ to emphasize that the operator driving the flow is constructed from the stress tensor of the deformed theory, at finite $\lambda$, rather than from the stress tensor $T_{\mu\nu}(0)$ of the undeformed theory $\mathcal{L}_0$. However, to lighten our notation, we will suppress the dependence on $\lambda$ and simply write $T_{\mu\nu}$ when it is clear from context which stress tensor is indicated. We also write $f(T)$ rather than $f(T_{\mu\nu})$ for short. Note that in the function $f(T_{\mu\nu})$ we might also allow dependence upon derivatives of the stress tensor. However, for simplicity, we neglect this option in the present work, though we believe the arguments below would generalise to this case too.

Now consider a scale transformation of the deformed theory $\mathcal{L}_\lambda$. Under an infinitesimal scale transformation $g_{\mu\nu} \to e^{2\epsilon} g_{\mu\nu} \simeq g_{\mu\nu} + 2\epsilon g_{\mu\nu}$, the change in the action $S_\lambda$ is

$$\delta S_\lambda = \frac{\delta S_\lambda}{\delta g^{\mu\nu}} \delta g^{\mu\nu} = -\epsilon \int d^d x \, \sqrt{-g} \, T^\mu{}_\mu \,, \tag{20}$$

where we have used the definition of the Hilbert stress tensor, $T_{\mu\nu} = -\frac{2}{\sqrt{-g}} \frac{\delta S}{\delta g^{\mu\nu}}$. Such a scale transformation dilates lengths by a factor of $e^\epsilon$ and thus diminishes mass scales by a factor of $e^{-\epsilon}$.

Because $\mathcal{L}_0$ is assumed to be conformally invariant, and thus there is no characteristic scale in the undeformed theory, the only scale in the deformed theory is the one set by $\lambda$. If $\lambda$ has length dimension $\Delta$, we can define an energy scale $\Lambda$ by

$$\lambda = \frac{1}{\Lambda^\Delta}. \tag{21}$$

For a theory with a single energy scale $\Lambda$, the effect of a scale transformation is identical to the effect of modifying this energy scale as $\Lambda \to e^{-\epsilon} \Lambda$ or $\log(\Lambda) \longrightarrow \log(\Lambda) - \epsilon$. Thus such a change in the energy scale is controlled by the trace of the Hilbert stress tensor as

$$\frac{dS_\lambda}{d\log(\Lambda)} = \Lambda \frac{dS_\lambda}{d\Lambda} = \int d^d x \, \sqrt{-g} \, T^\mu{}_\mu \,. \tag{22}$$

Although we have derived this relation using the Hilbert stress tensor, it also holds for other stress tensors obtained by an improvement transformation, since they differ by an on-shell total derivative which vanishes when integrated over spacetime as in (22).

On the other hand, we can rewrite the flow equation $\partial_\lambda \mathcal{L} = f(T)$ in terms of $\Lambda$:

$$\frac{\partial S_\lambda}{\partial \lambda} = -\frac{\Lambda^{\Delta+1}}{\Delta} \frac{\partial S_\lambda}{\partial \Lambda} = \int d^d x \, \sqrt{-g} \, f(T). \tag{23}$$

Comparing (22) and (23) and equating the integrands[4] we find that

$$-\frac{1}{\Delta}\Lambda^d T^\mu{}_\mu = f(T), \tag{24}$$

or in terms of the flow parameter $\lambda$,

$$T^\mu{}_\mu = -\Delta\lambda f(T) \iff \lambda\frac{\partial\mathcal{L}_\lambda}{\partial\lambda} = -\frac{1}{\Delta}T^\mu{}_\mu. \tag{25}$$

Equation (25) is the general trace flow equation for deformations by any scalar operator constructed from the stress tensor passing through a seed conformal field theory.[5]

Note that, as a consequence of this trace flow equation, any stress tensor deformation of a CFT can be rewritten in a form that is driven by the trace:

$$\frac{\partial\mathcal{L}_\lambda}{\partial\lambda} = f\left(T_{\mu\nu}(\lambda)\right) \iff \frac{\partial\mathcal{L}_\lambda}{\partial\lambda} = -\frac{T^\mu{}_\mu}{\lambda\Delta}. \tag{26}$$

One might be tempted to conclude that a generic stress tensor flow can therefore be replaced with a deformation by the trace. However, the equivalence (26) is misleading because the right side of the rightmost equation is indeterminate: both the numerator $T^\mu{}_\mu$ and denominator $\lambda\Delta$ are vanishing in the limit $\lambda \to 0$. Such a trace flow equation, therefore cannot correctly reproduce the deformation around a conformal seed theory $\mathcal{L}_0$. For this reason, although the trace flow equation is useful, we will view the deformation by the operator $f\left(T_{\mu\nu}(\lambda)\right)$ as the more fundamental one since it is well-defined as $\lambda \to 0$.

We will now specialize to the combination of interest, which is the $d$-dimensional $T^2$ operator with $\Delta = d$ which we study in the present work:

$$f(T) = O_{T^2} \equiv \frac{1}{2d}\left(T^{\mu\nu}T_{\mu\nu} - \frac{2}{d}\left(T^\mu{}_\mu\right)^2\right). \tag{27}$$

We stress that, at the quantum level, the combination $O_{T^2}$ only defines a local operator by point-splitting in $d = 2$. In the present work we will primarily restrict attention to classical flow equations for the Lagrangian, thinking of the object (27) as a combination of classical field variables rather than as a local operator in the spectrum of the theory. However, we note in passing that the trace flow equation is believed to hold at the quantum level for theories which arise as $T\bar{T}$ deformations of two-dimensional conformal field theories. In that context, one has the operator equation

$$T^\mu{}_\mu(x) = -2\lambda O_{T\bar{T}}(x), \tag{28}$$

where on the right side we write $O_{T\bar{T}}(x)$ rather than $O_{T^2}$ to emphasize that this object is now the local operator defined for $d = 2$ by

$$O_{T\bar{T}}(x) = \lim_{y\to x}\left(T^{\mu\nu}(x)T_{\mu\nu}(y) - T^\mu{}_\mu(x)T^\nu{}_\nu(y)\right). \tag{29}$$

In this two-dimensional setting, equation (28) holds as a relationship between operators inside of correlation functions which plays an important role in conformal perturbation theory in this context. Because this ingredient in our analysis can be promoted to a statement about the quantum theory, it would be interesting to investigate whether the arguments which we will present in the remainder of this section also have analogues at the quantum level. However, we will leave this question to future work and for the remainder of this paper we will focus on a purely classical analysis.

---

[4]The subsequent results hold only for the Hilbert stress tensor, and not for improvements thereof, since we no longer integrate over spacetime and thus total derivatives may contribute. Note also that there are scale-invariant theories for which the trace of the Hilbert stress tensor is an on-shell total derivative. These subtleties are important for the $3d$ stress tensor flows for scalar theories considered in [32].

[5]Although we have made no additional assumptions about the field content of the theory $\mathcal{L}_\lambda$, or any additional symmetries such as electric-magnetic duality invariance, we note that a relation of the form (25) can be derived via different means in the context of $4d$ duality-invariant electrodynamics [59, 60].

## 2.2 Relevant $T\overline{T}$-like flows

Now suppose that, as in Section 2.1, the Lagrangian $\mathcal{L}_\lambda$ solves the $T^2$ flow equation with an initial condition $\mathcal{L}_0$ that has no characteristic length scale. In particular, the stress tensor associated with $\mathcal{L}_\lambda$ satisfies the trace flow equation (25) with $f(T) = O_{T^2}$. We then define the "subtracted" theory

$$\widetilde{\mathcal{L}} = \mathcal{L} - \frac{1}{\lambda}. \tag{30}$$

The stress tensor $\widetilde{T}_{\mu\nu}$ of $\widetilde{\mathcal{L}}$ is related to that of $\mathcal{L}$ as

$$\widetilde{T}_{\mu\nu} = T_{\mu\nu} - \frac{1}{\lambda} g_{\mu\nu}. \tag{31}$$

One finds

$$\widetilde{T}^{\mu\nu}\widetilde{T}_{\mu\nu} = T^{\mu\nu}T_{\mu\nu} - \frac{2}{\lambda}T^\mu{}_\mu + \frac{d}{\lambda^2}, \qquad \widetilde{T}^\mu{}_\mu = T^\mu{}_\mu - \frac{d}{\lambda}, \tag{32}$$

and therefore the new $T^2$ operator for $\widetilde{\mathcal{L}}$ is

$$\widetilde{O}_{T^2} = \frac{1}{2d}\left( \widetilde{T}^{\mu\nu}\widetilde{T}_{\mu\nu} - \frac{2}{d}\left(\widetilde{T}^\mu{}_\mu\right)^2 \right)$$
$$= O_{T^2} + \frac{1}{\lambda d}T^\mu{}_\mu - \frac{1}{2\lambda^2}. \tag{33}$$

However, by the trace flow equation (25) with $f(T) = O_{T^2}$, we have

$$O_{T^2} + \frac{1}{\lambda d}T^\mu{}_\mu = 0, \tag{34}$$

so we conclude that

$$\widetilde{O}_{T^2} = -\frac{1}{2\lambda^2}. \tag{35}$$

That is, the $T^2$ operator for the subtracted theory $\widetilde{\mathcal{L}}_\lambda$ is actually a constant. We can use this to rewrite the flow equation for any such subtracted theory in a different way. Beginning from the form (25) with $f(T) = O_{T^2}$ of the flow equation for $\mathcal{L}_\lambda$, and making the replacements

$$\mathcal{L} = \widetilde{\mathcal{L}} + \frac{1}{\lambda}, \qquad T^\mu{}_\mu = \widetilde{T}^\mu{}_\mu + \frac{d}{\lambda}, \qquad \lambda = \frac{1}{\sqrt{2|\widetilde{O}_{T^2}|}}, \tag{36}$$

where we assume $\lambda > 0$, one finds

$$\lambda^2 \frac{\partial \widetilde{\mathcal{L}}}{\partial \lambda} = -\frac{1}{d}\frac{\widetilde{T}^\mu{}_\mu}{\sqrt{2|\widetilde{O}_{T^2}|}}. \tag{37}$$

Finally, shifting variables to $T = \frac{1}{\lambda}$ and substituting the definition of $\widetilde{O}_{T^2}$, we conclude

$$\frac{\partial \widetilde{\mathcal{L}}}{\partial T} = \frac{\widetilde{T}^\mu{}_\mu}{\sqrt{\left| d\widetilde{T}^{\rho\sigma}\widetilde{T}_{\rho\sigma} - 2\left(\widetilde{T}^\rho{}_\rho\right)^2 \right|}}. \tag{38}$$

Note that the combination on the right side of (38) is dimensionless, so that this is a flow equation driven by a relevant operator, unlike the conventional $T^2$ deformation which is defined in terms of an irrelevant operator.

A similar construction would have allowed us to write relevant flow equations for deformations by other quadratic combinations of stress tensors, such as $c_1 T^{\mu\nu} T_{\mu\nu} + c_2 \left(T^{\mu}{}_{\mu}\right)^2$, by defining a subtracted Lagrangian $\widetilde{\mathcal{L}} = \mathcal{L} - \frac{a}{\lambda}$ where these constants satisfy

$$ac_1 d + ac_2 d^2 = -\frac{1}{2}. \tag{39}$$

However, for simplicity, in this paper we will consider only the choice $c_1 = \frac{1}{2d}$, $c_2 = -\frac{1}{d^2}$, $a = 1$ which is presented above.

## 2.3 $T^2$ fixed points

The examples constructed in Section 2.2, whose $T^2$ operators are constants which are independent of fields, are special insofar as the equations of motion for such theories are invariant to leading order under an infinitesmal $T^2$ flow. This is obvious at first order in the deformation parameter $\lambda$, since by assumption the effect of the $T^2$ deformation is simply to add a constant term to the Lagrangian, which does not affect the dynamics.

However, beyond leading order, it is possible that additional structures will be generated and that the invariance of the equations of motion will fail. In this Subsection, we will demonstrate a sufficient condition for the invariance of the equations of motion to continue to hold at all orders in the deformation parameter.[6]

Let $\mathcal{L}$ be any Lagrangian in $d$ spacetime dimensions with the following two properties:

(i) The $T^2$ operator of such a theory is

$$O_{T^2} = c_1 \kappa^2, \tag{40}$$

where $c_1$ is dimensionless and $\kappa$ is a constant with mass dimension $d$.

(ii) The trace of the stress tensor is proportional to the undeformed Lagrangian itself,

$$T^{\mu}{}_{\mu} = c_2 \mathcal{L}, \tag{41}$$

for some other dimensionless constant $c_2$.

In Section 3, we will see that the Plebanski theory of electrodynamics [61] in four spacetime dimensions is an example which satisfies these two properties. We will also construct a new theory of scalars in $d = 2$ which falls into the same class of examples.

We now consider a $T^2$ deformation of a theory $\mathcal{L}$ which satisfies properties (i) - (ii). To leading order in the flow parameter $\lambda$, the deformed theory is

$$\mathcal{L}_{\lambda} = \mathcal{L}_0 + c_1 \lambda \kappa^2 + \mathcal{O}(\lambda^2). \tag{42}$$

We therefore make an ansatz for the all-orders Lagrangian which takes the form

$$\mathcal{L}_{\lambda} = f(\lambda\kappa)\mathcal{L}_0 + c_1 \kappa g(\lambda\kappa), \tag{43}$$

where $f$ and $g$ are functions of the dimensionless combination $\chi \equiv \lambda\kappa$ which satisfy the initial conditions

$$f(0) = 1, \qquad g(0) = 0. \tag{44}$$

---

[6]We state these conditions for the operator $O_{T^2}$ defined in (27) but an analogous argument can be given for deformations by other quadratic combinations of stress tensors.

The stress tensor $T_{\mu\nu}(\lambda)$ for the ansatz (43) is simply

$$T_{\mu\nu}(\lambda) = f(\chi)T_{\mu\nu}(0) + c_1\kappa g(\chi)g_{\mu\nu}, \tag{45}$$

where $T_{\mu\nu}(0)$ is the stress tensor of the undeformed theory $\mathcal{L}_0$. Therefore, one finds that the $T^2$ operator associated with our ansatz at finite $\lambda$ is

$$O_{T^2}(\lambda) = c_1\kappa^2 f(\chi)^2 - \frac{c_1}{d}\kappa f(\chi)g(\chi)T^\mu_{\ \mu}(0) - \frac{c_1^2\kappa^2}{2}g(\chi)^2, \tag{46}$$

where we have used the assumption that

$$O_{T^2}(0) = \frac{1}{2d}\left(T^{\mu\nu}(0)T_{\mu\nu}(0) - \frac{2}{d}\left(T^\mu_{\ \mu}(0)\right)^2\right) = c_1\kappa^2. \tag{47}$$

The differential equation $\partial_\lambda \mathcal{L}_\lambda = O_{T^2}(\lambda)$, which our ansatz (43) should satisfy, is

$$f'(\chi)\mathcal{L}_0 + c_1\kappa g'(\chi) = c_1\kappa\left(f(\chi)^2 - \frac{c_1}{2}g(\chi)^2\right) - \frac{c_1}{d}f(\chi)g(\chi)T^\mu_{\ \mu}(0). \tag{48}$$

We now see that, in order for our ansatz to be consistent, we must have that $T^\mu_{\ \mu}(0)$ be proportional to $\mathcal{L}_0$ in order to match the non-constant terms on either side of equation (48). When property (ii) is satisfied, our differential equation becomes

$$g'(\chi) = f(\chi)^2 - \frac{c_1}{2}g(\chi)^2, \qquad f'(\chi) = -\frac{c_1 c_2}{d}f(\chi)g(\chi). \tag{49}$$

The general solution to this system of ordinary differential equations can be written in terms of an implicit expression involving an unevaluated integral. We will focus on a special case where the resulting integral simplifies, namely

$$c_1 = 2, \qquad c_2 = d, \tag{50}$$

which are the values that will appear in our examples of Section 3. In this case, the solution is simply

$$f(\chi) = \frac{1}{1+\chi^2}, \qquad g(\chi) = \frac{\chi}{1+\chi^2}. \tag{51}$$

In particular, this implies that the full solution to the $T^2$ flow equation,

$$\mathcal{L}_\lambda = \frac{\mathcal{L}_0}{1+\kappa^2\lambda^2} + \frac{2\lambda\kappa^2}{1+\lambda^2\kappa^2}, \tag{52}$$

is merely a constant rescaling of the undeformed Lagrangian $\mathcal{L}_0$, along with a constant shift. Neither the additive constant nor the multiplicative prefactor affects the equations of motion for the model, so in this case, we see that the dynamics of the theory are invariant under a $T^2$ flow. We refer to such an invariant seed theory as a $T^2$ fixed point.

## 3 Theories related to zero-birefringence conditions

As an application of the formalism developed in Section 2, we will now study several examples of theories which are motivated by studies of zero-birefringence conditions. We focus on four-dimensional Abelian gauge theories, although we will discuss two-dimensional analogues of these theories which involve scalar fields in Section 4. We will find that each of the theories in this family is related to one of the analyses of the preceding Section, such as the subtracted $T\bar{T}$-like flows or $T^2$ fixed points.

As in the preceding sections, we stress that all of these results hold only for classical stress tensor deformations of the Lagrangian. We will not address the well-known issues that arise in attempting to define a quantum $T\bar{T}$ operator in spacetime dimensions $d > 2$, but see [62] for a discussion of these subtleties.

## 3.1 Compatibility of $T^2$ flow and zero-birefringence conditions

One of the motivations for studying the two-dimensional $T\overline{T}$ operator, and its higher-dimensional analogues, is that such deformations appear to preserve many symmetries and desirable properties of the seed theory. For instance, it is often possible to present such deformations as superspace flow equations, which makes it manifest that the deformation preserves the supersymmetry of the initial theory [10–13, 15–17]. In the $2d$ setting, it is also known that the $T\overline{T}$ deformation preserves the integrability of the seed theory [2].

It is natural to wonder whether similar stress tensor flow equations preserve other interesting properties of their seed theories. For $4d$ gauge theories, one physically motivated condition that one might impose is the absence of birefringence, or a polarization-dependent dispersion relation. The constraints on a theory to guarantee zero-birefringence is an old topic that has been studied by many authors; see [61, 63–68] for some of the original analysis or [69] for a recent discussion. For our purposes, the most convenient way of expressing this condition is as a pair of partial differential equations for theories of non-linear electrodynamics described by the Lagrangian $\mathcal{L}(S, P)$:

$$\mathcal{L}_S \left(\mathcal{L}_{SS} - \mathcal{L}_{PP}\right) = 2S \left(\mathcal{L}_{SS} \mathcal{L}_{PP} - \mathcal{L}_{SP}^2\right), \tag{53a}$$

$$\mathcal{L}_S \mathcal{L}_{SP} = P \left(\mathcal{L}_{SS} \mathcal{L}_{PP} - \mathcal{L}_{SP}^2\right). \tag{53b}$$

Here subscripts indicate partial derivatives with respect to the argument.

In this section, we will prove that the $T^2$ flow is the only irrelevant stress tensor deformation compatible with the zero-birefringence conditions (53). That is, if one begins with an initial theory $\mathcal{L}_0$ which exhibits no birefringence, and then constructs the one-parameter family of theories $\mathcal{L}_\lambda$ satisfying

$$\frac{\partial \mathcal{L}_\lambda}{\partial \lambda} = f\left(T_{\mu\nu}\right), \tag{54}$$

then the only choice of an irrelevant operator $f\left(T_{\mu\nu}\right)$ for which all of the theories $\mathcal{L}_\lambda$ will also satisfy the zero-birefringence conditions is

$$f\left(T_{\mu\nu}\right) = a\left(T^{\mu\nu} T_{\mu\nu} - \frac{1}{2}\left(T^\mu_{\ \mu}\right)^2\right). \tag{55}$$

This singles out the operator $O_{T^2}$ up to an overall proportionality constant $a$.

To show this, it is convenient to first compute $T_{\mu\nu}$ for a general Lagrangian of the form

$$\mathcal{L} = \mathcal{L}(S, P), \tag{56}$$

in four spacetime dimensions. The Hilbert stress tensor is given by

$$\begin{aligned} T_{\mu\nu} &= -2 \frac{\partial \mathcal{L}}{\partial g^{\mu\nu}} + g_{\mu\nu} \mathcal{L} \\ &= -2 \left(\frac{\partial \mathcal{L}}{\partial S} \frac{\partial S}{\partial g^{\mu\nu}} + \frac{\partial \mathcal{L}}{\partial P} \frac{\partial P}{\partial g^{\mu\nu}}\right) + g_{\mu\nu} \mathcal{L}, \end{aligned} \tag{57}$$

where

$$\frac{\partial S}{\partial g^{\mu\nu}} = -\frac{1}{2} F_\mu^{\ \rho} F_{\nu\rho}, \qquad \frac{\partial P}{\partial g^{\mu\nu}} = -\frac{1}{2} F_{(\mu}^{\ \ \rho} \widetilde{F}_{\nu)\rho}. \tag{58}$$

We can then compute the two Lorentz scalars

$$T^\mu_{\ \mu} = -4 \left(S \frac{\partial \mathcal{L}}{\partial S} + P \frac{\partial \mathcal{L}}{\partial P} - \mathcal{L}\right), \tag{59a}$$

$$T^{\mu\nu} T_{\mu\nu} = 4 \left(\left(\mathcal{L} - P \frac{\partial \mathcal{L}}{\partial P}\right)^2 - 2S \frac{\partial \mathcal{L}}{\partial S} \left(\mathcal{L} - P \frac{\partial \mathcal{L}}{\partial P}\right) + (P^2 + 2S^2) \left(\frac{\partial \mathcal{L}}{\partial S}\right)^2\right). \tag{59b}$$

To obtain the expressions (59), one must use various identities relating the traces of powers of $4 \times 4$ matrices. We refer the reader to [16] or [49] for details on this procedure.

Using (59), one can construct the four-dimensional $T^2$ operator,

$$
\begin{aligned}
O_{T^2} &= \frac{1}{8}\left( T^{\mu\nu}T_{\mu\nu} - \frac{1}{2}\left(T^{\mu}{}_{\mu}\right)^2 \right) \\
&= -\frac{1}{2}\left( \mathcal{L} - P\frac{\partial \mathcal{L}}{\partial P} \right)^2 + S\left( \mathcal{L} - P\frac{\partial \mathcal{L}}{\partial P} \right)\frac{\partial \mathcal{L}}{\partial S} + \frac{1}{2}P^2\left(\frac{\partial \mathcal{L}}{\partial S}\right)^2.
\end{aligned}
\tag{60}
$$

Beginning from a seed Lagrangian $\mathcal{L}(S, P)$, a continuous deformation by the operator $O_{T^2}$ defines a flow in the space of field theories described by the differential equation

$$
\frac{\partial \mathcal{L}(\lambda, S, P)}{\partial \lambda} = -\frac{1}{2}\left[ \left( \mathcal{L} - P\frac{\partial \mathcal{L}}{\partial P} \right)^2 - 2S\left( \mathcal{L} - P\frac{\partial \mathcal{L}}{\partial P} \right)\frac{\partial \mathcal{L}}{\partial S} - P^2\left(\frac{\partial \mathcal{L}}{\partial S}\right)^2 \right].
\tag{61}
$$

We will now find the criteria under which a stress tensor deformation preserves the zero-birefringence conditions, and check that the flow equation (61) satisfies these criteria. First consider a one-parameter family of Lagrangians $\mathcal{L}(\lambda, S, P)$ which obeys a flow equation driven by a general Lorentz scalar constructed from the stress tensor,

$$
\frac{\partial \mathcal{L}}{\partial \lambda} = f\left( T^{\mu}{}_{\mu}, T^{\mu\nu}T_{\mu\nu} \right),
\tag{62}
$$

where $f$ is an arbitrary[7] function. To ease notation, we will define

$$
y_1 = T^{\mu}{}_{\mu}, \qquad y_2 = T^{\mu\nu}T_{\mu\nu}.
\tag{63}
$$

We would like to impose the condition that the entire family of Lagrangians $\mathcal{L}(\lambda, S, P)$ all satisfy the zero-birefringence constraints (53) at any value of $\lambda$. In particular, we may differentiate the two conditions (53) with respect to $\lambda$ to obtain

$$
\begin{aligned}
f_S\left(\mathcal{L}_{SS} - \mathcal{L}_{PP}\right) + \mathcal{L}_S\left(f_{SS} - f_{PP}\right) &= 2S\left(f_{SS}\mathcal{L}_{PP} + \mathcal{L}_{SS}f_{PP} - 2f_{SP}\mathcal{L}_{SP}\right), \\
f_S\mathcal{L}_{SP} + \mathcal{L}_S f_{SP} &= P\left(f_{SS}\mathcal{L}_{PP} + \mathcal{L}_{SS}f_{PP} - 2f_{SP}\mathcal{L}_{SP}\right).
\end{aligned}
\tag{64}
$$

Here $f_S = \frac{\partial f}{\partial S}$ and $f_P = \frac{\partial f}{\partial P}$. We can re-express these conditions in terms of $f_{y_1} = \frac{\partial f}{\partial y_1}$ and $f_{y_2} = \frac{\partial f}{\partial y_2}$ using the expressions (59), which gives

$$
\begin{aligned}
f_S &= -4f_{y_1}\left(P\mathcal{L}_{SP} + S\mathcal{L}_{SS}\right) + 8f_{y_2}\Big[ S\mathcal{L}_S^2 + (P\mathcal{L}_P - \mathcal{L})(P\mathcal{L}_{SP} + S\mathcal{L}_{SS}) \\
&\quad + \mathcal{L}_S\left(SP\mathcal{L}_{SP} + (P^2 + 2S^2)\mathcal{L}_{SS}\right) \Big], \\
f_P &= -4f_{y_1}\left(P\mathcal{L}_{PP} + S\mathcal{L}_{SP}\right) + 8f_{y_2}\Big[ P\mathcal{L}_P(P\mathcal{L}_{PP} + S\mathcal{L}_{SP}) - \mathcal{L}(P\mathcal{L}_{PP} + S\mathcal{L}_{SP}) \\
&\quad + \mathcal{L}_S\left(PS\mathcal{L}_{PP} + P\mathcal{L}_S + (P^2 + 2S^2)\mathcal{L}_{SP}\right) \Big],
\end{aligned}
\tag{65}
$$

and similar (but more cumbersome) expressions for $f_{SS}, f_{SP}$, and $f_{PP}$. We substitute each of these expressions for the partial derivatives of $f$ into (64), simplify by assuming that the Lagrangian $\mathcal{L}$ satisfies the original pair of conditions (53), and then collect all terms proportional to each independent derivative of $\mathcal{L}$. For instance, the coefficient multiplying $\mathcal{L}_S$ must vanish

---

[7]The attentive reader may wonder why we have only allowed the function $f$ to depend on the two traces $y_1 = \mathrm{Tr}(T)$ and $y_2 = \mathrm{Tr}(T^2)$ but not on $\mathrm{Tr}(T^3)$ or $\mathrm{Tr}(T^4)$. Although a general $4 \times 4$ matrix $T$ has four independent traces, the stress tensor associated with an Abelian gauge theory in four dimensions has only two independent eigenvalues, as shown in Appendix A of [49]; as a result, it also has only two independent traces. Thus it suffices to consider dependence on only two invariants.

independently, as must the coefficient multiplying $\mathcal{L}_{SP}$, and so on. After doing this, one finds that the two conditions (64) are both satisfied if and only if

$$f_{y_2 y_2} = f_{y_1 y_2} = 0, \qquad f_{y_1 y_1} = -f_{y_2}. \tag{66}$$

Thus the deforming operator $f(y_1, y_2)$ must be at most linear in $y_2 = T^{\mu\nu}T_{\mu\nu}$ and at most quadratic in $y_1 = T^\mu{}_\mu$. Furthermore, it must have a relative coefficient of $-\frac{1}{2}$ between the $y_2$ term and the $y_1^2$ term. The most general function which satisfies these properties is

$$f(T_{\mu\nu}) = a\left(T^{\mu\nu}T_{\mu\nu} - \frac{1}{2}\left(T^\mu{}_\mu\right)^2\right) + b\,T^\mu{}_\mu + c\,, \tag{67}$$

where $a, b, c$ and constants independent of $T_{\mu\nu}$. The third term is merely a constant shift which has no effect on the equations of motion. The second term is a deformation proportional to the trace of the stress tensor, which generates scale transformations; this is permissible because the property of exhibiting zero-birefringence is scale-invariant.[8] We also point out that generic deformations of a conformal seed theory can be recast in the form of the second term in (67) due to the trace flow equation, but following the remarks around equation (26), this form of the deformation is not valid as $\lambda \to 0$.

Ignoring constant shifts and scale transformations, the only non-trivial deformation which satisfies our conditions is then

$$f(T_{\mu\nu}) = a\left(T^{\mu\nu}T_{\mu\nu} - \frac{1}{2}\left(T^\mu{}_\mu\right)^2\right) \sim O_{T^2}\,. \tag{68}$$

This argument can be seen as a different way of motivating the particular deforming operator $O_{T^2}$, with the specific relative coefficient of $-\frac{1}{2}$, as this choice is the only irrelevant stress tensor deformation which is compatible with the zero-birefringence condition.

### 3.2 Examples of zero-birefringence theories in 4d electrodynamics

We have just shown by a general argument that the four-dimensional $T^2$ flow equation preserves the zero-birefringence conditions, at least to first order. However, it is also illuminating to study the flow in explicit examples of theories which satisfy this condition. In fact, such a study can be carried out in an exhaustive case-by-case manner, since it was recently shown in [69] that there are only three theories of non-linear electrodynamics in four spacetime dimensions which can be written in terms of a Lagrangian density $\mathcal{L}(S, P)$ and which satisfy the zero-birefringence condition. We recall from (11) that $S$ and $P$ are the two independent Lorentz invariants which can be constructed from $F_{\mu\nu}$, namely

$$S = -\frac{1}{4}F_{\mu\nu}F^{\mu\nu}, \qquad P = -\frac{1}{4}F_{\mu\nu}\widetilde{F}^{\mu\nu}. \tag{69}$$

The theories of electrodynamics with no birefringence fall into three classes:

(I) The conventional Born-Infeld theory, whose Lagrangian can be written as

$$\mathcal{L}_{\mathrm{BI}} = T - \sqrt{T^2 - 2TS - P^2}\,, \tag{70}$$

where $T$ is a dimensionful parameter with the interpretation of a $D3$-brane tension.

---

[8]Note that birefringence is the property that an unpolarized beam of light passing through a background field will split into two separate beams with a relative angle between them. Conformal transformations distort lengths but not angles, so the presence of birefringence is conformally invariant.

(II) The theory of Plebanski electrodynamics, with action

$$\mathcal{L}_{\text{Pl}} = \frac{\kappa S}{P}, \tag{71}$$

and where $\kappa$ is another dimensionful constant.

(III) The so-called "reverse Born-Infeld" theory with Lagrangian

$$\mathcal{L}_{\text{rBI}} = \alpha\sqrt{P^2 + 2TS - T^2} + \beta P. \tag{72}$$

There is also a fourth theory satisfying the zero-birefringence condition which is referred to in [69] as "extreme Born-Infeld" or eBI. However, the eBI theory does not admit a description in terms of a conventional Lagrangian density $\mathcal{L}(S, P)$, but rather as a Lagrangian constraint relating the variables $S$ and $P$. As a result, we will not consider this theory in the present work.

We have reviewed that the standard Born-Infeld Lagrangian (70) is the solution to a stress tensor flow equation given in equation (12). One might ask whether the other two solutions (71), (72) to the zero-birefringence condition also satisfy some flow equation. We will see that the answer is yes in both cases, up to a rescaling of the Lagrangian and addition of a constant term in the case of the Plebanski theory.

This is perhaps expected in the case of the reverse-Born-Infeld theory, since the Lagrangian (72) is related to the usual Born-Infeld Lagrangian (70) by dropping the constant $T$ term, adding a term proportional to $P$, and reversing a sign under the square root. Interpreting the tension $T$ as $\frac{1}{\lambda}$, we see that the step of dropping the constant term is identical to the subtraction procedure which was studied in Section 2.2. Indeed, we will find that the reverse Born-Infeld Lagrangian satisfies the flow equation which we derived for such subtracted theories. By adding back this term, and formally continuing certain real parameters in the solution to complex values, one can also show that a version of the reverse Born-Infeld theory satisfies a conventional $T\overline{T}$-like flow equation but with an imaginary value of the flow parameter $\lambda$.

Although it is less obvious whether the Plebanski Lagrangian might satisfy any version of a $T\overline{T}$-like flow, it will turn out that this theory is exactly one of the $T^2$ fixed points which we considered in Section 2.3. In this sense, the Plebanski theory is something of an edge case, since the equations of motion of this model are left invariant under the $T^2$ deformation: to all orders in $\lambda$, the effect of the flow is merely to re-scale the Lagrangian by an overall prefactor and add a constant shift which does not affect the dynamics.

### *Born-Infeld*

The most well-known of the three Lagrangian solutions to the zero-birefringence condition is the Born-Infeld theory. We saw above that the Born-Infeld Lagrangian (70), written in terms of $\lambda = \frac{1}{T}$ as

$$\mathcal{L}_{\text{BI}} = \frac{1}{\lambda}\left(1 - \sqrt{1 - 2\lambda S - \lambda^2 P^2}\right), \tag{73}$$

is the solution to the classical $T\overline{T}$-like flow equation (12), as shown in [34]. The supersymmetric version of this theory also satisfies a manifestly supersymmetric flow equation in superspace [15] – we will elaborate on other supersymmetric flows in Section 5.

The initial condition for this flow equation is

$$\lim_{\lambda \to 0} \mathcal{L}_{\text{BI}} = S = -\frac{1}{4}F_{\mu\nu}F^{\mu\nu}, \tag{74}$$

which is the usual Maxwell Lagrangian. In particular, this is a conformally invariant seed theory, which means that the only scale in the deformed theory is the one set by $\lambda$, and the analysis

of Section 2.1 implies that the stress tensor of the deformed theory $\mathcal{L}_{\text{BI}}$ satisfies the trace flow equation. Following the general arguments of Section 2.2, we may therefore considered the subtracted version of the Born-Infeld Lagrangian,

$$\widetilde{\mathcal{L}}_{\text{BI}} = \mathcal{L}_{\text{BI}} - \frac{1}{\lambda} = -\frac{1}{\lambda}\sqrt{1 - 2\lambda S - \lambda^2 P^2}\,. \tag{75}$$

The Lagrangian (75) then satisfies a flow equation driven by a relevant operator,

$$\frac{\partial \widetilde{\mathcal{L}}_{\text{BI}}}{\partial T} = \frac{\widetilde{T}^{\mu}{}_{\mu}}{\sqrt{\left|4\widetilde{T}^{\rho\sigma}\widetilde{T}_{\rho\sigma} - 2\left(\widetilde{T}^{\rho}{}_{\rho}\right)^2\right|}}\,. \tag{76}$$

Here $\widetilde{T}^{\mu\nu}$ is the stress tensor associated with $\widetilde{\mathcal{L}}_{\text{BI}}$ and we assume that $T = \frac{1}{\lambda} > 0$. Thus the Born-Infeld Lagrangian, without constant term, is an example of the theories considered in Section 2. Our general arguments imply that $\widetilde{\mathcal{L}}_{\text{BI}}$ has the unusual feature that the quadratic combination of stress tensors which usually drives the $T^2$ flow is constant:

$$\widetilde{T}^{\mu\nu}\widetilde{T}_{\mu\nu} - \frac{1}{2}\left(\widetilde{T}^{\mu}{}_{\mu}\right)^2 = -4T^2\,, \tag{77}$$

which means that the absolute value under the square root in the flow equation (76) picks out the positive combination $\left|\widetilde{T}^{\mu\nu}\widetilde{T}_{\mu\nu} - \frac{1}{2}\left(\widetilde{T}^{\mu}{}_{\mu}\right)^2\right| = 4T^2$.

This relevant flow equation may initially seem to be a contradiction to the general argument of Section 3.1, since the subtracted theories $\widetilde{\mathcal{L}}_{\text{BI}}$ still exhibit zero birefringence at any value of $T$, and yet the flow equation (76) appears to be driven by an operator which is not of the form (67). However, it is important to note that the combination $\widetilde{O}_{T^2} = \widetilde{T}^{\rho\sigma}\widetilde{T}_{\rho\sigma} - \frac{1}{2}\left(\widetilde{T}^{\rho}{}_{\rho}\right)^2$ is actually a constant for this class of theories. Therefore, the flow equation (76) is really of the form $\frac{\partial \widetilde{\mathcal{L}}_{\text{BI}}}{\partial T} = b\widetilde{T}^{\mu}{}_{\mu}$ for constant $b$, which is indeed compatible with the general solution (67) with $a = c = 0$.

We also note that the flow equation continues to hold if we recale the Born-Infeld Lagrangian or add any term which does not contribute to the stress tensor. For instance, if we instead define

$$\widetilde{\mathcal{L}}_{\text{BI}} \longrightarrow \widetilde{\mathcal{L}}'_{\text{BI}} = \alpha\widetilde{\mathcal{L}}_{\text{BI}} + \beta P\,, \tag{78}$$

then the new theory satisfies

$$\frac{\partial \widetilde{\mathcal{L}}'_{\text{BI}}}{\partial T} = \frac{\alpha\widetilde{T}^{\mu}{}_{\mu}}{\sqrt{\left|4\widetilde{T}^{\rho\sigma}\widetilde{T}_{\rho\sigma} - 2\left(\widetilde{T}^{\rho}{}_{\rho}\right)^2\right|}}\,, \tag{79}$$

because $P$ is a total derivative which does not couple to the metric. This will be useful when we consider the reverse Born-Infeld theory shortly, which can be interpreted as a member of the rescaled class of theories (78) for imaginary $\alpha$.

### *Plebanski*

We now consider the second solution to the zero-birefringence condition, which we refer to as the Plebanski theory. In equation (71) we have written the Lagrangian for this theory as $\mathcal{L}_{\text{Pl}} = \frac{\kappa S}{P}$. However, in order to match conventions with the general discussion of Section 2.3, it is convenient to rescale[9] the value of $\kappa$ in the Plebanski Lagrangian to

$$\mathcal{L}_{\text{Pl}} = \frac{2\kappa S}{P}\,. \tag{80}$$

---

[9]Alternatively, one could repeat the general analysis of Section 2.3 and solve the coupled differential equations with $c_1 = \frac{1}{2}$ rather than $c_1 = 2$, which leads to the same rescaling.

This Plebanski Lagrangian shares the property of the subtracted Born-Infeld Lagrangian $\widetilde{\mathcal{L}}_{\mathrm{BI}}$ that a particular combination of stress tensor bilinears yields a constant. By evaluating the stress tensor contractions for $\mathcal{L}_{\mathrm{Pl}}$ using the general formula (59), one finds

$$T^{\mu}{}_{\mu} = \frac{8\kappa S}{P}, \qquad T^{\mu\nu}T_{\mu\nu} = 16\kappa^2\left(1 + \frac{2S^2}{P^2}\right) = \frac{1}{2}\left(T^{\mu}{}_{\mu}\right)^2 + 16\kappa^2. \tag{81}$$

The $T^2$ operator for this theory is therefore

$$O_{T^2} = \frac{1}{8}\left(T^{\mu\nu}T_{\mu\nu} - \frac{1}{2}\left(T^{\mu}{}_{\mu}\right)^2\right) = 2\kappa^2. \tag{82}$$

In the notation of equations (40) - (41), we see that the Plebanski theory satisfies

$$O_{T^2} = c_1\kappa^2, \qquad T^{\mu}{}_{\mu} = c_2\mathcal{L}_{\mathrm{Pl}}, \tag{83}$$

with $c_1 = 2$ and $c_2 = 4$. We can then quote the full solution to the $T^2$ flow equation with initial condition $\mathcal{L}_0 = \mathcal{L}_{\mathrm{Pl}}$ which is

$$\mathcal{L}_\lambda = \frac{\mathcal{L}_{\mathrm{Pl}}}{1 + \kappa^2\lambda^2} + \frac{2\lambda\kappa^2}{1 + \kappa^2\lambda^2}. \tag{84}$$

The second term is a field-independent constant which does not affect the equations of motion, whereas the first term is simply an overall rescaling of the undeformed Lagrangian.

We therefore see that the one-parameter family of Plebanski theories, labeled by the parameter $\kappa$, is closed under $T^2$ flows: up to an additive constant, the effect of a $T^2$ deformation is simply to rescale

$$\kappa \longrightarrow \frac{\kappa}{1 + \lambda^2\kappa^2}, \tag{85}$$

while remaining within the same class of theories. Because the parameter $\kappa$ drops out of the equations of motion for this model, one can view this family of Lagrangians as defining a single physical theory regardless of the value of $\kappa$. From this perspective, the theory is genuinely invariant under the $T^2$ flow.

### Reverse Born-Infeld

We now turn our attention to the more unusual case of reverse Born-Infeld electrodynamics, described by the Lagrangian

$$\mathcal{L}_{\mathrm{rBI}} = \alpha\sqrt{P^2 + 2TS - T^2} + \beta P, \tag{86}$$

where $\alpha$ and $\beta$ are dimensionless constants.

Whereas the usual Born-Infeld theory exhibits a maximum allowed value for the electric field, since the magnitude of the electric field vector $\vec{E}$ must be bounded above in order for the argument of the square root to remain positive, the reverse Born-Infeld theory instead has a *minimum* allowed electric field: the usual inequality on $|\vec{E}|$ is reversed to give a *lower* bound. It is straightforward to find these bounds using

$$S = \frac{1}{2}\left(|\vec{E}|^2 - |\vec{B}|^2\right), \qquad P = \vec{E}\cdot\vec{B}, \tag{87}$$

where $\vec{E}, \vec{B}$ are the three-vector electric and magnetic fields, respectively. In terms of these three-vectors and the tension $T = \frac{1}{\lambda}$, the usual Born-Infeld Lagrangian is then

$$\mathcal{L}_{\mathrm{BI}} = T - \sqrt{T^2 - T\left(|\vec{E}|^2 - |\vec{B}|^2\right) - \left(\vec{E}\cdot\vec{B}\right)^2}. \tag{88}$$

One can bound the argument of the square root by applying the Cauchy-Schwarz inequality to $\left(\vec{E}\cdot\vec{B}\right)^2$. In order for this argument to remain positive for any $\vec{B}$, we require

$$\left|\vec{E}\right|^2 < T\,. \tag{89}$$

On the other hand, the reverse Born-Infeld Lagrangian is

$$\mathcal{L}_{\mathrm{rBI}} = \alpha\sqrt{\left(\vec{E}\cdot\vec{B}\right)^2 + T\left(\left|\vec{E}\right|^2 - \left|\vec{B}\right|^2\right) - T^2} + \beta\left(\vec{E}\cdot\vec{B}\right)\,, \tag{90}$$

so it is instead necessary (but not sufficient) for reality that

$$\left|\vec{E}\right|^2 > T\,. \tag{91}$$

To see that this bound is not sufficient, note that one can choose a large magnetic field $\vec{B}$ which is orthogonal to $\vec{E}$ so that $\left(\vec{E}\cdot\vec{B}\right)^2 - T\left|\vec{B}\right|^2$ is large and negative. The constraint (91) is the claimed lower bound on the magnitude of the electric field in the reverse Born-Infeld theory, which is the opposite of the usual inequality (89).

The Lagrangian $\mathcal{L}_{\mathrm{rBI}}$ satisfies flow equations similar to those of the usual Born-Infeld Lagrangian, although the signs of several quantities will be reversed. First we note that this theory satisfies the same flow equation (79) as for the rescaled Born-Infeld theory,

$$\frac{\partial \mathcal{L}_{\mathrm{rBI}}}{\partial T} = \frac{\alpha T^{\mu}{}_{\mu}}{\sqrt{\left|4T^{\rho\sigma}T_{\rho\sigma} - 2\left(T^{\rho}{}_{\rho}\right)^2\right|}}\,, \tag{92}$$

for any value of $\alpha$ and $\beta$. Therefore the reverse Born-Infeld theory obeys the same flow, driven by the same relevant operator constructed from stress tensors, as the Born-Infeld theory with the constant term subtracted. From the perspective of the flow equation, the only difference in the reverse Born-Infeld case is that the combination of stress tensors in the denominator now takes a positive constant value:

$$T^{\mu\nu}T_{\mu\nu} - \frac{1}{2}\left(T^{\mu}{}_{\mu}\right)^2 = 4\alpha^2 T^2\,. \tag{93}$$

This reflects the fact that, if one neglects the $\beta P$ term appearing in the reverse Born-Infeld Lagrangian (whose contribution drops out of $T^{\mu}{}_{\mu}$ and $T^{\mu\nu}T_{\mu\nu}$), the Lagrangian $\mathcal{L}_{\mathrm{rBI}}$ reduces to the usual Born-Infeld Lagrangian $\mathcal{L}_{\mathrm{BI}}$ if we set $\alpha = \mathrm{i}$.

This suggests that the Born-Infeld and reverse Born-Infeld theories belong to a single family of Lagrangians which are related by formal analytic continuation of certain coupling constants. In fact, we can show that this entire family of theories satisfies a version of the $\lambda$-flow equation driven by $O_{T^2}$ discussed above, if we formally allow the flow parameter $\lambda$ to become complex.

To show this, it is convenient to first rewrite the reverse Born-Infeld Lagrangian in a form which has a finite weak-field ($T \to \infty$) limit by adding an imaginary constant:

$$\begin{aligned}\mathcal{L}_{\mathrm{rBI}} &= \alpha\sqrt{P^2 + 2TS - T^2} + \beta P - \mathrm{i}\alpha T \\ &= \beta P - \mathrm{i}\alpha S - \frac{\mathrm{i}\alpha}{2T}\left(S^2 + P^2\right) + \mathcal{O}\left(\frac{1}{T^2}\right)\,. \end{aligned} \tag{94}$$

At large tension, this Lagrangian reduces to the Maxwell Lagrangian $S$ multiplied by an imaginary constant, along with a total derivative term $\beta P$.

One can express this Lagrangian in terms of $\lambda = \frac{1}{T}$ to find

$$\mathcal{L}_{\mathrm{rBI}} = \frac{\alpha}{\lambda}\left(\sqrt{-1 + 2\lambda S + \lambda^2 P^2} - \mathrm{i}\right) + \beta P\,. \tag{95}$$

This Lagrangian is now of the same schematic form as the usual Born-Infeld Lagrangian, before performing the subtraction procedure. Accordingly, it satisfies the flow equation

$$\frac{\partial \mathcal{L}_{\text{rBI}}}{\partial \lambda} = \frac{\mathrm{i}}{8\alpha} \left( T^{\mu\nu} T_{\mu\nu} - \frac{1}{2} \left( T^{\mu}{}_{\mu} \right)^2 \right)$$
$$= \frac{\mathrm{i}}{\alpha} O_{T^2} , \tag{96}$$

for any value of $\alpha$ and $\beta$. We can interpret the differential equation (96) as a formal analytic continuation of the usual $T^2$ flow for the Born-Infeld action to complex values of the parameters. At small $\lambda$, the Lagrangian (95) approaches

$$\mathcal{L}_{\text{rBI}} = -\mathrm{i}\alpha S + \beta P + \mathcal{O}(\lambda), \tag{97}$$

whereas at large $\lambda$ it approaches

$$\mathcal{L}_{\text{rBI}} = (\alpha + \beta) P + \mathcal{O}\left( \frac{1}{\lambda} \right). \tag{98}$$

Therefore, we can view the reverse Born-Infeld Lagrangian (with the addition of the imaginary constant term) as solving a $T^2$ flow equation with either of the initial conditions (97), (98) at $\lambda \to 0$ or $\lambda \to \infty$, respectively.

In fact, there is an entire $U(1)$'s worth of such theories: for any angle $\theta$, the theory

$$\mathcal{L}_{\theta \text{BI}} = \frac{\alpha e^{\mathrm{i}\theta}}{\lambda} \left( 1 - \sqrt{1 - 2\lambda S - \lambda^2 P^2} \right) + \beta P , \tag{99}$$

can be shown to satisfy the flow equation

$$\frac{\partial \mathcal{L}_{\theta \text{BI}}}{\partial \lambda} = \frac{e^{-\mathrm{i}\theta}}{8\alpha} \left( T^{\mu\nu} T_{\mu\nu} - \frac{1}{2} \left( T^{\mu}{}_{\mu} \right)^2 \right)$$
$$= \frac{e^{-\mathrm{i}\theta}}{\alpha} O_{T^2} . \tag{100}$$

The case when $\alpha = 1$ and $\theta = 0$ corresponds to the ordinary Born-Infeld theory, whereas $\theta = -\frac{\pi}{2}$ recovers the reverse Born-Infeld theory which we considered above. Formally speaking, one can recast the general flow equation for any $\theta$ by defining a complex number

$$z = \alpha e^{\mathrm{i}\theta} , \tag{101}$$

and a new complex flow parameter

$$\widehat{\lambda} = \frac{\lambda}{z} , \tag{102}$$

so that the flow equation can be written as

$$\frac{\partial \mathcal{L}_{\theta \text{BI}}}{\partial \widehat{\lambda}} = O_{T^2} . \tag{103}$$

We may therefore interpret this entire family of theories as solving a $T^2$ flow equation where the flow parameter $\widehat{\lambda}$ is now complex. Alternatively, one can generate any theory in this family by beginning with the Born-Infeld-like solution for real $\lambda$,

$$\mathcal{L}_{\text{BI}} = \frac{\alpha}{\lambda} \left( 1 - \sqrt{1 - 2\lambda S - \lambda^2 P^2} \right) + \beta P , \tag{104}$$

and simultaneously making the replacements

$$\lambda \to \lambda e^{-\mathrm{i}\theta} , \quad S \to S e^{\mathrm{i}\theta} , \quad P \to P e^{\mathrm{i}\theta} , \quad \beta \to \beta e^{-\mathrm{i}\theta} . \tag{105}$$

This collection of replacements yields the version of Born-Infeld in equation (99) at arbitrary $\theta$. Of course, this Lagrangian $\mathcal{L}_{\theta\mathrm{BI}}$ can formally be viewed as a holomorphic function of complex variables $\lambda, S, P, \beta \in \mathbb{C}$. From this perspective, the observation that the family of theories satisfies the flow equation (103) is a consequence of holomorphicity in $\lambda$, since one can differentiate along any direction in the complex $\lambda$ plane.

Finally, we point out one example which *cannot* be realized as a stress tensor flow of this form. So far in this discussion, we have treated $\alpha$ and $\beta$ as arbitrary dimensionless constants. However, in [69] the authors considered a case in which these constants are chosen in a way which is correlated with the tension parameter $T$. If we take

$$\alpha = \beta = \frac{\kappa}{T}, \tag{106}$$

where $\kappa$ is a new constant with the same dimensions as the tension $T$, then the reverse Born-Infeld Lagrangian can be written as

$$\mathcal{L}_{\mathrm{rBI}} = \frac{\kappa}{T} \left( \sqrt{P^2 + 2TS - T^2} - P \right), \tag{107}$$

in terms of the tension $T = \frac{1}{\lambda}$. The theory described by (107) cannot be realized as a $T\overline{T}$-like flow in the way that we have been discussing. One way to see this is to note that, in the limit as $T \to 0$, this theory reduces to the Plebanski theory,

$$\mathcal{L}_{\mathrm{rBI}} = \frac{\kappa S}{P} + \mathcal{O}(T). \tag{108}$$

We have seen above that the Plebanski Lagrangian is a fixed point of the four-dimensional $T\overline{T}$-like flow. Because the choice of parameters for the reverse Born-Infeld theory appearing in (107) reduces to a fixed point of the $O_{T^2}$ deformation in a particular limit, we conclude that it cannot be realized as an irrelevant flow beginning from an initial condition near $T = 0$.

### 3.3 Properties of root-$T^2$ flows

In the previous Subsection, we have considered irrelevant flows for theories of electrodynamics – or in some cases, relevant flows with an inverted coupling constant – which are driven by combinations built from the energy-momentum tensor. However, one can also study deformations by *marginal* operators constructed from $T_{\mu\nu}$. One example in this class is the root-$T\overline{T}$ operator in two spacetime dimensions [35] and its higher-dimensional generalizations. In general dimension $d$, we define the root-$T^2$ operator $R$ as

$$R = \sqrt{\frac{1}{d} T^{\mu\nu} T_{\mu\nu} - \frac{1}{d^2} \left( T^{\mu}{}_{\mu} \right)^2}. \tag{109}$$

Note that the operator above can be expressed as

$$R = \frac{1}{\sqrt{d}} \sqrt{\hat{T}^{\mu\nu} \hat{T}_{\mu\nu}}, \qquad \hat{T}_{\mu\nu} := T_{\mu\nu} - \frac{1}{d} g_{\mu\nu} T^{\rho}{}_{\rho}, \tag{110}$$

which makes explicit the fact that the traceless part of the stress energy tensor is associated to this flow. In the context of $4d$ theories, the operator $R$ is

$$R = \frac{1}{2} \sqrt{T^{\mu\nu} T_{\mu\nu} - \frac{1}{4} \left( T^{\mu}{}_{\mu} \right)^2}, \tag{111}$$

and for a general Lagrangian $\mathcal{L}(S,P)$ for a $4d$ Abelian gauge theory, the combination appearing under the square root in equation (111) takes a special form:

$$T^{\mu\nu}T_{\mu\nu} - \frac{1}{4}\left(T^{\mu}{}_{\mu}\right)^2 = 4(S^2 + P^2)\left(\frac{\partial\mathcal{L}}{\partial S}\right)^2 . \tag{112}$$

Therefore, assuming[10] that $\frac{\partial\mathcal{L}}{\partial S} > 0$, the root-$T^2$ operator can be written simply as

$$R = \sqrt{S^2 + P^2}\,\frac{\partial\mathcal{L}}{\partial S} . \tag{113}$$

Unlike the operator $O_{T^2}$ considered above, it is straightforward to check that a flow equation driven by the operator $R$ is *not* compatible with the zero-birefringence condition. That is, beginning with a theory $\mathcal{L}$ which satisfies (53) and deforming it according to $\frac{\partial\mathcal{L}}{\partial\gamma} = R$ produces a theory which no longer satisfies (53) – indeed, this must have been the case, since we have shown in Section 3.1 that $O_{T^2}$ is the unique stress tensor deformation which preserves this condition. Therefore, deforming a theory of $4d$ electrodynamics by root-$T^2$ does not preserve the property of exhibiting zero-birefringence.

However, there are other properties of a theory which are preserved by the root-$T^2$ flow. One example is conformal invariance: as one would expect for a classically marginal deformation, if the trace of the stress tensor vanishes for the seed theory $\mathcal{L}_0$, then the trace of the stress tensor for the deformed theory $\mathcal{L}_\gamma$ also vanishes.

This claim is simple to verify. Using equation (59), the trace of the stress tensor associated with a general Lagrangian $\mathcal{L}(S,P)$ is

$$T^{\mu}{}_{\mu} = -4\left(S\frac{\partial\mathcal{L}}{\partial S} + P\frac{\partial\mathcal{L}}{\partial P} - \mathcal{L}\right), \tag{114}$$

so if the family of Lagrangians $\mathcal{L}(S,P,\gamma)$ satisfies a flow equation of the form

$$\frac{\partial\mathcal{L}}{\partial\gamma} = f\left(T_{\mu\nu}\right), \tag{115}$$

where $f$ is some Lorentz scalar constructed from the stress tensor, then one finds

$$\frac{\partial}{\partial\gamma}T^{\mu}{}_{\mu} = -4\left(S\frac{\partial f}{\partial S} + P\frac{\partial f}{\partial P} - f\right). \tag{116}$$

Therefore, the trace of the stress tensor will not flow (to leading order) so long as

$$f = S\frac{\partial f}{\partial S} + P\frac{\partial f}{\partial P} . \tag{117}$$

The most general Lorentz scalar which depends on the stress tensor is a function of the two invariants $y_1 = T^{\mu}{}_{\mu}, y_2 = T^{\mu\nu}T_{\mu\nu}$, as we have used in Section 3.1. We now wish to find the constraints on the function $f(y_1, y_2)$ such that, if the trace $T^{\mu}{}_{\mu}$ of the stress tensor vanishes, then the derivative $\partial_\gamma T^{\mu}{}_{\mu}$ also vanishes. Since $T^{\mu}{}_{\mu} = 0$, we have

$$\mathcal{L} = S\frac{\partial\mathcal{L}}{\partial S} + P\frac{\partial\mathcal{L}}{\partial P} . \tag{118}$$

Using the partial derivatives $f_S, f_P$ computed in (65) above, substituting into (117), and using the assumption (118) and its derivatives, one finds that $\partial_\gamma T^{\mu}{}_{\mu} = 0$ if

$$\left(f - 2y_2\partial_{y_2}f\right)\big|_{y_1=0} = 0, \tag{119}$$

---

[10]If instead $\frac{\partial\mathcal{L}}{\partial S} < 0$, this can be absorbed into a redefinition of the flow parameter as $\gamma \to -\gamma$.

which is satisfied by

$$f(y_1, y_2) = R = \frac{1}{2}\sqrt{y_2 - \frac{1}{2}y_1^2}, \tag{120}$$

or more generally this condition holds for any function $f(y_1, y_2)$ which is proportional to $\sqrt{y_2}$ after setting $y_1 = 0$.

A more direct way to see that a deformation by the marginal combination $R$ preserves conformal invariance is to note that the general solution to the differential equation

$$\frac{\partial \mathcal{L}}{\partial \gamma} = \sqrt{S^2 + P^2}\,\frac{\partial \mathcal{L}}{\partial S}, \tag{121}$$

with initial condition $\mathcal{L}|_{\gamma=0} = \mathcal{L}_0(S, P)$ is given by

$$\mathcal{L}_\gamma = \mathcal{L}_0\left(\cosh(\gamma)S + \sinh(\gamma)\sqrt{S^2 + P^2},\, P\right). \tag{122}$$

That is, to solve the root-$T^2$ flow equation, one simply replaces $S$ with $\widetilde{S} = \cosh(\gamma)S + \sinh(\gamma)\sqrt{S^2 + P^2}$ everywhere in the undeformed Lagrangian $\mathcal{L}_0$. However, this change of variables has the property that

$$S\frac{\partial \mathcal{L}(S, P)}{\partial S} + P\frac{\partial \mathcal{L}(S, P)}{\partial P} = \widetilde{S}\frac{\partial \mathcal{L}(\widetilde{S}, P)}{\partial \widetilde{S}} + P\frac{\partial \mathcal{L}(\widetilde{S}, P)}{\partial P}. \tag{123}$$

Therefore, if the stress tensor $T^\mu{}_\mu(0)$ of the undeformed Lagrangian $\mathcal{L}_0(S, P)$ vanishes, so

$$S\frac{\partial \mathcal{L}_0}{\partial S} + P\frac{\partial \mathcal{L}_0}{\partial P} - \mathcal{L}_0 = 0, \tag{124}$$

then the stress tensor $T^\mu{}_\mu(\gamma)$ for the deformed Lagrangian $\mathcal{L}_\gamma$ also satisfies

$$\begin{aligned}
T^\mu{}_\mu(\gamma) &= S\frac{\partial \mathcal{L}_\gamma}{\partial S} + P\frac{\partial \mathcal{L}_\gamma}{\partial P} - \mathcal{L}_\gamma \\
&= \widetilde{S}\frac{\partial \mathcal{L}_\gamma}{\partial \widetilde{S}} + P\frac{\partial \mathcal{L}_\gamma}{\partial P} - \mathcal{L}_\gamma \\
&= \left(S\frac{\partial \mathcal{L}_0}{\partial S} + P\frac{\partial \mathcal{L}_0}{\partial P} - \mathcal{L}_0\right)\Big|_{S \to \widetilde{S}} \\
&= 0,
\end{aligned} \tag{125}$$

which confirms that the deformed stress tensor remains traceless. We reiterate that these arguments only establish that the *classical* root-$T^2$ deformation of the Lagrangian preserves conformal invariance of the corresponding classical field theory. It is not at all obvious that this statement can be lifted to an observation about the quantum theory, and we will not investigate this question here.

Another property of interest in theories of non-linear electrodynamics is electric-magnetic duality symmetry. For instance, the ModMax theory is special because it is the only conformally invariant extension of Maxwell theory which remains invariant under electric-magnetic duality rotations, and it is known that the ModMax theory is obtained from the Maxwell theory by a root-$T^2$ flow. One might therefore wonder whether the property of electric-magnetic duality invariance is preserved more generally by the root-$T^2$ deformation or by the $T^2$ deformation. In fact, this property is preserved by *any* deformation constructed from the stress tensor. We will now pause to demonstrate this fact to leading order in the deformation parameter; the extension of this argument to all orders can be found in [70].

Like the study of zero-birefringence conditions, invariance under electric-magnetic duality rotations is an old subject; see [71–80] and references therein for previous studies of conditions for duality invariance. The Euler-Lagrange equations associated with a Lagrangian $\mathcal{L}(S, P)$ respect electric-magnetic duality rotations if

$$\mathcal{L}_S^2 - \frac{2S}{P}\mathcal{L}_S\mathcal{L}_P - \mathcal{L}_P^2 = 1 \,. \tag{126}$$

We note that this condition is weaker than imposing that the Lagrangian itself be invariant under electric-magnetic duality; for instance, the free Maxwell Lagrangian $\mathcal{L} = S$ is not itself invariant under duality rotations, but its equations of motion are.

We claim that, if $\mathcal{L}$ is any Lagrangian satisfying the electric-magnetic duality condition (126), and if the deformed Lagrangian $\mathcal{L}_\gamma$ satisfies

$$\frac{\partial \mathcal{L}}{\partial \gamma} = f\left(T_{\mu\nu}\right), \tag{127}$$

then the deformed Lagrangian $\mathcal{L}_\gamma$ also satisfies this condition. We will check this by taking the derivative of (126) with respect to $\gamma$, which gives

$$\mathcal{L}_S f_S - \frac{S}{P}(f_S\mathcal{L}_P + \mathcal{L}_S f_P) - \mathcal{L}_P f_P = 0 \,. \tag{128}$$

We again assume that the function $f$ depends on $S, P$ only through the combinations $y_1 = T^\mu_{\ \mu}$, $y_2 = T^{\mu\nu}T_{\mu\nu}$, and substitute the partial derivatives (65). After doing this, simplifying the result using the relation (126) and its derivatives with respect to $S$ and $P$, and doing some algebra, we find that the constraint (128) holds *identically*, without any additional assumptions on the function $f(y_1, y_2)$. We conclude from this simple check that a theory with electric-magnetic duality invariance retains this property under any stress tensor deformation, such as root-$T^2$.

The argument we have just presented gives one proof that electric-magnetic duality is preserved to first order by explicitly using properties of the flow equation. However, note that an alternative and intuitive way of seeing this invariance is to note that, if the Euler-Lagrange equations associated with a theory are invariant under electric-magnetic duality, then the stress tensor $T_{\mu\nu}$ of such theory is also invariant; see [71] or section 2.1 of the lectures [81] for a proof. It then follows that any deformation constructed from the stress tensor preserves electric-magnetic duality symmetry.

## 3.4 Examples of root-$T^2$-deformed theories in 4d electrodynamics

In this Subsection, we will investigate the flows driven by the root-$T^2$ operator $R$ and whose seed theories correspond to each of the zero-birefringence theories in Section 3.2. Each of the resulting theories that we find can be interpreted as a two-parameter family of doubly-deformed theories, with one parameter $\lambda$ associated with the $T^2$ flow and a second parameter $\gamma$ associated with the root-$T^2$ flow. In fact, in all of these cases the flows actually commute, which is indicated schematically by the following diagram.

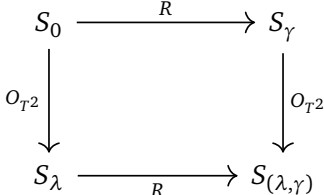

That is, in all three cases, one may either

(A) first deform the seed action $S_0$ by $O_{T^2}$ to obtain $S_\lambda$, and then deform the result by $R$ to find $S_{(\lambda,\gamma)}$, or

(B) first deform $S_0$ by $R$ to get $S_\gamma$, and then deform this theory by $O_{T^2}$ to obtain $S_{(\lambda,\gamma)}$,

and the results of procedures (A) and (B) agree.

### *ModMax-Born-Infeld*

The result of performing a root-$T^2$ deformation whose initial condition is the Born-Infeld theory is the ModMax-Born-Infeld theory, which was first written down in [45]. This is a two-parameter collection of theories, labeled by $\lambda$ and $\gamma$, which reduces to the Born-Infeld theory in the limit $\gamma \to 0$. The other limit $\lambda \to 0$ of these models yields the Modified Maxwell or ModMax theory of [44]. The interpretation of this family of theories in terms of stress tensor flows has already been investigated in [38, 48, 49], so here we will only briefly review these results and point out that this theory also satisfies a relevant flow equation of the form discussed in Section 2.

In our normalization, the ModMax-Born-Infeld Lagrangian is

$$\mathcal{L}_{\gamma\mathrm{BI}} = \frac{1}{\lambda}\left(1 - \sqrt{1 - 2\lambda\left(\cosh(\gamma)S + \sinh(\gamma)\sqrt{S^2 + P^2}\right) - \lambda^2 P^2}\right). \tag{129}$$

The model (129) is electric-magnetic duality invariant, and thus satisfies the differential equation (126), but does not satisfy the zero-birefringence condition (53) when $\gamma \neq 0$. This is expected from the observation that deformations by $O_{T^2}$, but not by the root-$T^2$ operator $R$, preserves the zero-birefringence constraint.

The ModMax-Born-Infeld Lagrangian satisfies the two commuting flow equations

$$\frac{\partial \mathcal{L}_{\gamma\mathrm{BI}}}{\partial \lambda} = O_{T^2}, \qquad \frac{\partial \mathcal{L}_{\gamma\mathrm{BI}}}{\partial \gamma} = R, \tag{130}$$

where $R$ takes the form of equation (113) appropriate for $4d$ gauge theories.

Because these flow equations commute, we can interpret $\mathcal{L}_{\gamma\mathrm{BI}}$ either as a root-$T^2$ deformation of the Born-Infeld theory or as a $T^2$ deformation of the ModMax theory. Since the ModMax theory is conformally invariant, the latter interpretation as a $T^2$ deformation of a conformally invariant seed theory suggests – by the general analysis of Section 2 – that the subtracted version of this Lagrangian should also satisfy a flow equation driven by a relevant operator. Indeed this is the case. If we define

$$\widetilde{\mathcal{L}}_{\gamma\mathrm{BI}} = \mathcal{L}_{\gamma\mathrm{BI}} - \frac{1}{\lambda}$$
$$= -\frac{1}{\lambda}\sqrt{1 - 2\lambda\left(\cosh(\gamma)S + \sinh(\gamma)\sqrt{S^2 + P^2}\right) - \lambda^2 P^2}, \tag{131}$$

and set $T = \frac{1}{\lambda}$, then one can verify that this Lagrangian obeys the flow

$$\frac{\partial \widetilde{\mathcal{L}}_{\gamma\mathrm{BI}}}{\partial T} = \frac{\widetilde{T}^\mu{}_\mu}{\sqrt{\left|4\widetilde{T}^{\rho\sigma}\widetilde{T}_{\rho\sigma} - 2\left(\widetilde{T}^\rho{}_\rho\right)^2\right|}}, \tag{132}$$

where $\widetilde{T}_{\mu\nu}$ is the stress tensor associated with $\widetilde{\mathcal{L}}_{\gamma\mathrm{BI}}$. This is the same relevant flow equation obeyed by the ordinary Born-Infeld theory.

*Modified Plebanski*

Although we have seen in Section 3.2 that the Plebanski theory is a fixed point of the $T^2$ flow, which is related to the fact that the quadratic combination $O_{T^2}$ of stress tensors reduces to a constant for this theory, the combination appearing in the operator $R$ is *not* a constant. Therefore the root-$T^2$ flow

$$\frac{\partial \mathcal{L}}{\partial \gamma} = \frac{1}{2}\sqrt{T^{\mu\nu}T_{\mu\nu} - \frac{1}{4}\left(T^{\mu}{}_{\mu}\right)^2}\,, \tag{133}$$

with initial condition

$$\mathcal{L}\big|_{\gamma=0} = \frac{\kappa S}{P} = \mathcal{L}_{\text{Pl}}\,, \tag{134}$$

will lead to a non-trivial modification of the theory. Note that, if the seed theory in the flow (133) were the Maxwell Lagrangian, the solution to this flow equation would be the ModMax theory. In this case, with the Plebanski Lagrangian as the initial condition, the solution to the flow equation is instead

$$\mathcal{L}_{\text{mPl}} = \cosh(\gamma)\frac{\kappa S}{P} + \kappa \sinh(\gamma)\sqrt{1 + \frac{S^2}{P^2}}\,. \tag{135}$$

We refer to this as the "modified Plebanski" theory. Note that

$$\mathcal{L}_{\text{mPl}} = \frac{\kappa}{P}\mathcal{L}_{\text{ModMax}}\,, \tag{136}$$

where

$$\mathcal{L}_{\text{ModMax}} = \cosh(\gamma)S + \sinh(\gamma)\sqrt{S^2 + P^2}\,, \tag{137}$$

which means that the $\gamma$-flow "commutes with division by $P$" in the sense that the solution to the flow equation (133) with initial condition $\mathcal{L}_{\text{Maxwell}} = S$ is the ModMax Lagrangian $\mathcal{L}_{\text{ModMax}}$, while the solution to the same flow equation with initial condition $\mathcal{L}_{\text{Pl}} = \frac{\kappa}{P}\mathcal{L}_{\text{Maxwell}}$ is $\mathcal{L}_{\text{mPl}} = \frac{\kappa}{P}\mathcal{L}_{\text{ModMax}}$. This is expected from the general solution (122) to the root-$T^2$ flow equation, which instructs us to replace $S$ with $\widetilde{S} = \mathcal{L}_{\text{ModMax}}$ in the undeformed Lagrangian $\mathcal{L}_0$, but leave the dependence on $P$ unchanged.

Because $T\overline{T}$-type flows and root-$T\overline{T}$-type flows commute, and since the Plebanski Lagrangian is a fixed point of the usual $T\overline{T}$-like flow (up to re-scaling and addition of a constant), one might think that the modified Plebanski theory (136) is also a fixed point of the $T\overline{T}$ deformation in the same sense. This is indeed the case. To see this, it is again convenient to first re-scale $\kappa$ by a factor of 2 to write

$$\mathcal{L}_{\text{mPL}} = \cosh(\gamma)\frac{2\kappa S}{P} + 2\kappa \sinh(\gamma)\sqrt{1 + \frac{S^2}{P^2}}\,. \tag{138}$$

One finds that the stress tensor $T_{\mu\nu}$ associated with (138) satisfies

$$\frac{1}{8}\left(T^{\mu\nu}T_{\mu\nu} - \frac{1}{2}\left(T^{\mu}{}_{\mu}\right)^2\right) = 2\kappa^2\,, \qquad T^{\mu}{}_{\mu} = 4\mathcal{L}_{\text{mPL}}\,. \tag{139}$$

The (rescaled) modified Plebanski theory therefore falls into the same class of theories which we considered in Section 2.3, so that the full solution to the $T^2$ flow equation is

$$\mathcal{L}_{\lambda} = \frac{\mathcal{L}_{\text{mPL}}}{1 + \kappa^2\lambda^2} + \frac{2\lambda\kappa^2}{1 + \kappa^2\lambda^2}\,. \tag{140}$$

Thus the modified Plebanski Lagrangian is also unaffected by the $T\overline{T}$ flow, up to the addition of a constant and an overall rescaling which do not affect the equations of motion, exactly as in the case of the usual Plebanski theory.

*ModMax-Reverse-Born-Infeld*

It is also possible to extend the reverse Born-Infeld theory to include a ModMax-like dependence, precisely as we have discussed for the ordinary Born-Infeld theory. Consider

$$\mathcal{L}_{\gamma\mathrm{rBI}} = \frac{\alpha}{\lambda}\left(\sqrt{-1 + 2\lambda\left(\cosh(\gamma)S + \sinh(\gamma)\sqrt{S^2 + P^2}\right) + \lambda^2 P^2} - \mathrm{i}\right) + \beta P, \tag{141}$$

which reduces to the reverse Born-Infeld Lagrangian (95) as $\gamma \to 0$. This two-parameter family of Lagrangians satisfies a similar pair of commuting flow equations,

$$\frac{\partial \mathcal{L}_{\gamma\mathrm{rBI}}}{\partial \lambda} = \frac{\mathrm{i}}{\alpha} O_{T^2}, \qquad \frac{\partial \mathcal{L}_{\gamma\mathrm{rBI}}}{\partial \gamma} = R, \tag{142}$$

at any value of $\lambda$ and $\gamma$.

All of the same comments about interpreting these theories in terms of complex values of the parameters, which we described around equation (99) for the $\gamma = 0$ limit of these theories, also apply to $\mathcal{L}_{\gamma\mathrm{rBI}}$ at finite $\gamma$. Explicitly, for any angle $\theta$ we can consider

$$\mathcal{L}_{(\theta,\gamma)\mathrm{BI}} = \frac{\alpha e^{\mathrm{i}\theta}}{\lambda}\left(1 - \sqrt{1 - 2\lambda\left(\cosh(\gamma)S + \sinh(\gamma)\sqrt{S^2 + P^2}\right) - \lambda^2 P^2}\right) + \beta P, \tag{143}$$

which satisfies

$$\frac{\partial \mathcal{L}_{(\theta,\gamma)\mathrm{BI}}}{\partial \lambda} = \frac{e^{-\mathrm{i}\theta}}{8\alpha}\left(T^{\mu\nu}T_{\mu\nu} - \frac{1}{2}\left(T^{\mu}{}_{\mu}\right)^2\right) = \frac{e^{-\mathrm{i}\theta}}{\alpha} O_{T^2}, \tag{144}$$

and this flow equation can be viewed as an holomorphic $T^2$ flow with a complex value of the deformation parameter $\lambda$.

Likewise, the subtracted form[11] of any member of this family of theories, defined as

$$\widetilde{\mathcal{L}}_{(\theta,\gamma)\mathrm{BI}} = \mathcal{L}_{(\theta,\gamma)\mathrm{BI}} - \frac{\alpha e^{\mathrm{i}\theta}}{\lambda}, \tag{145}$$

satisfies the expected relevant flow equation in terms of $T = \frac{1}{\lambda}$,

$$\frac{\partial \widetilde{\mathcal{L}}_{(\theta,\gamma)\mathrm{BI}}}{\partial T} = \frac{\alpha \widetilde{T}^{\mu}{}_{\mu}}{\sqrt{\left|4\widetilde{T}^{\rho\sigma}\widetilde{T}_{\rho\sigma} - 2\left(\widetilde{T}^{\rho}{}_{\rho}\right)^2\right|}}. \tag{146}$$

To conclude this section, we also mention that, purely from the formal point of view of the flow equations, one could not only analytically continue $\lambda$ but also the Root-$T\overline{T}$-like flow parameter $\gamma$. For instance, a purely imaginary $\gamma$ would turn the hyperbolic functions in (137) into $\sin(\gamma)$ and $\cos(\gamma)$ while keeping the structure of $\mathcal{L}_{\mathrm{ModMax}}$ preserved.

## 4 Scalar analogues in 2*d*

We now consider theories of a collection of scalar fields $\phi^i$, $i = 1,\ldots,N$, in two spacetime dimensions. These theories satisfy analogues of the flow equations for gauge theories discussed in the preceding section, and indeed the 2*d* scalar analogues can be obtained from the 4*d* theories by dimensional reduction. For instance, the dimensional reduction of the 4*d* ModMax theory to obtain a 2*d* Modified Scalar theory was performed in [38].

---

[11]Here we abuse notation somewhat, since we previously defined the "subtracted" form of a theory as $\widetilde{\mathcal{L}} = \mathcal{L} - \frac{1}{\lambda}$. Here the appropriate constant to subtract is instead $\frac{\alpha e^{\mathrm{i}\theta}}{\lambda}$, or $\widehat{\lambda}^{-1}$ in the notation of (102).

First it will be convenient to collect some general results before specializing to particular cases. Following the notation of [35], we first introduce the $2 \times 2$ matrix

$$X_{\mu\nu} = G_{ij}(\phi)\partial_\mu \phi^i \partial_\nu \phi^j, \tag{147}$$

where $i = 1, \ldots, N$ enumerates the scalars and $G_{ij}(\phi)$ is a target-space metric. A general $O(N)$ invariant Lagrangian for these $N$ scalar fields can depend only on the two independent traces

$$x_1 = \text{tr}(X) = X^\mu{}_\mu, \qquad x_2 = \text{tr}(X^2) = X^\mu{}_\nu X^\nu{}_\mu. \tag{148}$$

All higher invariants, such as $x_3 = \text{tr}(X^3)$, $x_4 = \text{tr}(X^4)$, and so on, will be related to the quantities $x_1$ and $x_2$ by trace identities. Any $O(N)$ invariant Lagrangian $\mathcal{L}(x_1, x_2)$, based on the $O(N)$ invariant building block $X_{\mu\nu}$, is therefore a function of the two independent traces $x_1, x_2$, much as a general Lagrangian for a $U(1)$ gauge theory in four dimensions constructed only from terms of powers of the field strength $F_{\mu\nu}$ is a function of the two invariants $S = -\frac{1}{4}F_{\mu\nu}F^{\mu\nu}$ and $P = -\frac{1}{4}F_{\mu\nu}\widetilde{F}^{\mu\nu}$.

The stress tensor $T_{\mu\nu}$ associated with such a general Lagrangian $\mathcal{L}(x_1, x_2)$ is given by

$$T_{\mu\nu} = -2X_{\mu\nu}\frac{\partial \mathcal{L}}{\partial x_1} - 4X_{\mu\nu}^2 \frac{\partial \mathcal{L}}{\partial x_2} + g_{\mu\nu}\mathcal{L}, \tag{149}$$

with $X_{\mu\nu}$ as in (147) and $X_{\mu\nu}^2 = X_\mu{}^\rho X_{\rho\mu}$. The two Lorentz scalars which we will need for constructing flows are

$$
\begin{aligned}
T^{\mu\nu}T_{\mu\nu} = {}& 2\left(\mathcal{L} + 2x_1^2 \frac{\partial \mathcal{L}}{\partial x_2}\right)\left(\mathcal{L} - 2x_1\left(\frac{\partial \mathcal{L}}{\partial x_1} + x_1 \frac{\partial \mathcal{L}}{\partial x_2}\right)\right) + 8x_2^2\left(\frac{\partial \mathcal{L}}{\partial x_2}\right)^2 \\
& + 4x_2\left(\left(\frac{\partial \mathcal{L}}{\partial x_1}\right)^2 + 6x_1\frac{\partial \mathcal{L}}{\partial x_1}\frac{\partial \mathcal{L}}{\partial x_2} - 2\frac{\partial \mathcal{L}}{\partial x_2}\left(\mathcal{L} - 2x_1^2\frac{\partial \mathcal{L}}{\partial x_2}\right)\right),
\end{aligned} \tag{150a}
$$

$$T^\mu{}_\mu = -2x_1 \frac{\partial \mathcal{L}}{\partial x_1} - 4x_2 \frac{\partial \mathcal{L}}{\partial x_2} + 2\mathcal{L}. \tag{150b}$$

There is a close analogy between the structure of theories $\mathcal{L}(S, P)$ of electrodynamics in four spacetime dimensions and scalar theories $\mathcal{L}(x_1, x_2)$ in two spacetime dimensions. Often one can map between the two classes of theories using the dictionary

$$S \longleftrightarrow x_1, \qquad P \longleftrightarrow \sqrt{2x_2 - 2x_1^2}. \tag{151}$$

For instance, one can write analogues of the zero-birefringence and electric-magnetic duality conditions for Lagrangians in $4d$ gauge theory, but for scalar theories in two dimensions, in terms of derivatives of $\mathcal{L}(x_1, x_2)$. The scalar versions of the two zero-birefringence condition (53) are

$$
\begin{aligned}
0 = {}& 2x_1\mathcal{L}_{x_2}^2 + 2\mathcal{L}_{x_1}\mathcal{L}_{x_2 x_2}\left(x_2 - 3x_1^2\right) - \mathcal{L}_{x_2}\left(\mathcal{L}_{x_1} + 12\mathcal{L}_{x_2 x_2}x_1\left(x_1^2 - x_2\right)\right) \\
& + 4x_1\mathcal{L}_{x_1 x_2}\left((x_1^2 - x_2)\mathcal{L}_{x_1 x_2} - \mathcal{L}_{x_1}\right) - \mathcal{L}_{x_1 x_1}\left(\mathcal{L}_{x_1} + 4\mathcal{L}_{x_2 x_2}x_1\left(x_1^2 - x_2\right)\right),
\end{aligned} \tag{152a}
$$

$$
\begin{aligned}
0 = {}& 2\mathcal{L}_{x_2}^2 + \mathcal{L}_{x_1 x_2}\left(2(x_1^2 - x_2)\mathcal{L}_{x_1 x_2} - \mathcal{L}_{x_1}\right) - 2\mathcal{L}_{x_2 x_2}\left(x_1\mathcal{L}_{x_1} + \left(x_1^2 - x_2\right)\mathcal{L}_{x_1 x_1}\right) \\
& + \mathcal{L}_{x_2}\left(4\mathcal{L}_{x_2 x_2}(x_2 - x_1^2) + \mathcal{L}_{x_1 x_1} + 2x_1\mathcal{L}_{x_1 x_2}\right).
\end{aligned} \tag{152b}
$$

As before, subscripts indicate partial derivatives with respect to the argument, so $\mathcal{L}_{x_i} = \frac{\partial \mathcal{L}}{\partial x_i}$. Although the conditions (152) appear more complicated than the analogous constraints in the $4d$ gauge theory case, and the connection to a physical condition such as the absence of birefringence is less clear, one can still show that the only stress tensor deformation (up to

an overall proportionality factor) which is compatible with these two differential equations is $O_{T^2} = \frac{1}{4}\left(T^{\mu\nu}T_{\mu\nu} - \left(T^\mu{}_\mu\right)^2\right)$. The derivation of this result is completely analogous to the $4d$ case of Section 3.1 and we do not repeat such a derivation for the $2d$ case. Similarly, the scalar version of the electric-magnetic duality invariance condition (126) is

$$\mathcal{L}_{x_1}^2 + 2x_1\mathcal{L}_{x_1}\mathcal{L}_{x_2} + 2(x_1^2 - x_2)\mathcal{L}_{x_2}^2 = 1. \tag{153}$$

The differential equation (153) is also compatible with *any* deformation by a Lorentz scalar constructed from the stress tensor, which follows from a calculation totally analogous to that of Section 3.3.[12] For instance, this condition is preserved under the root-$T\overline{T}$ deformation by the appropriate $2d$ version of the operator $R$, which we normalize as

$$R^{(2d)} = \sqrt{\frac{1}{2}T^{\mu\nu}T_{\mu\nu} - \frac{1}{4}\left(T^\mu{}_\mu\right)^2}. \tag{154}$$

As $R^{(2d)}$ is classically marginal, deforming a conformal field theory with this operator yields a deformed theory for which the trace of the stress tensor vanishes, as shown explicitly in [35]. This again mirrors the gauge theory result in Section 3.3.

In the following, we will consider several examples where performing the replacements (151) yields theories of scalars in $2d$ which satisfy similar $T^2$ and root-$T^2$ flow equations as the corresponding gauge theories in $4d$.

### *Modified-Nambu-Goto*

The scalar analogue of the ModMax-Born-Infeld theory, which was already considered in [35, 39], can be written as

$$\mathcal{L}_{s\gamma BI} = \frac{1}{\lambda}\left(1 - \sqrt{1 - \lambda x_1^{(\gamma)} + \frac{1}{2}\lambda^2\left(\left(x_1^{(\gamma)}\right)^2 - x_2^{(\gamma)}\right)}\right), \tag{155}$$

where we have defined

$$x_1^{(\gamma)} = \cosh(\gamma)x_1 + \sinh(\gamma)\sqrt{2x_2 - x_1^2}, \tag{156a}$$

$$x_2^{(\gamma)} = \cosh(2\gamma)x_2 + \sinh(2\gamma)x_1\sqrt{2x_2 - x_1^2}. \tag{156b}$$

This Lagrangian simultaneously satisfies the two flow equations

$$\frac{\partial\mathcal{L}_{s\gamma BI}}{\partial\lambda} = O_{T^2} = \frac{1}{4}\left(T^{\mu\nu}T_{\mu\nu} - \left(T^\mu{}_\mu\right)^2\right), \tag{157a}$$

$$\frac{\partial\mathcal{L}_{s\gamma BI}}{\partial\gamma} = R = \sqrt{\frac{1}{2}T^{\mu\nu}T_{\mu\nu} - \frac{1}{4}\left(T^\mu{}_\mu\right)^2}, \tag{157b}$$

as one can verify by evaluating the general expressions (150) for $\mathcal{L}_{s\gamma BI}$. The Modified-Nambu-Goto theory also satisfies the scalar zero-birefringence conditions (152) and the scalar electric-magnetic duality condition (153).

The subtracted version of this theory is

$$\widetilde{\mathcal{L}}_{s\gamma BI} = \mathcal{L}_{s\gamma BI} - \frac{1}{\lambda} = -\frac{1}{\lambda}\sqrt{1 - \lambda x_1^{(\gamma)} + \frac{1}{2}\lambda^2\left(\left(x_1^{(\gamma)}\right)^2 - x_2^{(\gamma)}\right)}, \tag{158}$$

---

[12]We also note that the same differential equation (153) appears in [43] as an integrability condition for theories related to the principal chiral model (PCM). For any Lagrangian which satisfies this condition, the equations of motion are equivalent to the flatness of a Lax connection which takes a particularly simple form. As this condition is preserved by any stress tensor deformation, any Lorentz scalar function $f\left(T_{\mu\nu}\right)$ can be used to build a classically integrable deformation of the PCM.

and based on the general arguments of Section 2.2, this Lagrangian obeys a differential equation driven by a relevant operator,

$$\frac{\partial \widetilde{\mathcal{L}}_{\text{s}\gamma\text{BI}}}{\partial T} = \frac{\widetilde{T}^{\mu}{}_{\mu}}{\sqrt{\left| 2\widetilde{T}^{\mu\nu}\widetilde{T}_{\mu\nu} - 2\left(\widetilde{T}^{\mu}{}_{\mu}\right)^2 \right|}}, \tag{159}$$

where $T = \frac{1}{\lambda}$ and $\widetilde{T}_{\mu\nu}$ is the stress tensor associated with $\widetilde{\mathcal{L}}_{\text{s}\gamma\text{BI}}$.

### *(Modified) scalar Plebanski*

The $4d$ Plebanski theory described by the Lagrangian

$$\mathcal{L}_{\text{Pl}} = \frac{\kappa S}{P}, \tag{160}$$

which is one of the three theories of electrodynamics satisfying the zero-birefringence condition which we studied in Section 3.2, also has a scalar analogue in two-dimensional field theory. Consider the theory defined by the Lagrangian

$$\mathcal{L}_{\text{sPl}} = \frac{\sqrt{2}\kappa x_1}{\sqrt{x_2 - x_1^2}}. \tag{161}$$

Here $\kappa$ is a constant with mass dimension 2 and the subscript "sPl" indicates "scalar Plebanski." We will assume that $x_2 > x_1^2$ so that the Lagrangian is real. The choice of normalization, with a factor of $\sqrt{2}$ in the numerator, is for later convenience. The scalar Plebanski Lagrangian $\mathcal{L}_{\text{sPl}}$ satisfies the scalar zero-birefringence conditions (152) but not the scalar electric-magnetic duality condition (153).

Both the four-dimensional Plebanski theory and its two-dimensional analogue share the property that they are, in a certain sense, fixed points of the appropriate $T\overline{T}$ flow. That is, for both theories, the effect of deforming the classical Lagrangian by the $T\overline{T}$ operator is simply an overall re-scaling of the kinetic term along with the addition of an unimportant constant. One can see this by computing the stress tensor associated with the scalar Plebanski Lagrangian (161) and appealing to the arguments of Section 2.3. The appropriate contractions of the stress tensor associated with (161) are

$$O_{T^2} = \frac{1}{4}\left( T^{\mu\nu}T_{\mu\nu} - \left(T^{\mu}{}_{\mu}\right)^2 \right) = 2\kappa^2, \qquad T^{\mu}{}_{\mu} = \frac{2\sqrt{2}x_1\kappa}{\sqrt{x_2 - x_1^2}} = 2\mathcal{L}_{\text{sPl}}. \tag{162}$$

We see that the scalar Plebanski theory shares the property (81) of the four-dimensional Plebanski theory, namely that the two Lorentz scalars $T^{\mu\nu}T_{\mu\nu}$ and $\left(T^{\mu}{}_{\mu}\right)^2$ that can be constructed from its stress tensor are dependent. In particular, using the notation of Section 2.3, this theory satisfies

$$O_{T^2} = c_1\kappa^2, \qquad T^{\mu}{}_{\mu} = c_2\mathcal{L}_{\text{sPl}}, \tag{163}$$

with $c_1 = 2$ and $c_2 = 2$. We can therefore invoke our previous general arguments about such theories to write down the full solution to the $T^2$ flow,

$$\mathcal{L}(\lambda) = \frac{\mathcal{L}_{\text{sPl}}}{1 + \kappa^2\lambda^2} + \frac{2\lambda\kappa^2}{1 + \lambda^2\kappa^2}. \tag{164}$$

This deformed Lagrangian has exactly the same structure as the solution (84) to the flow equation for the $4d$ Plebanski Lagrangian deformed by the appropriate $4d$ $T^2$ operator. Ignoring

the additive constant in (164), we see that the effect of the deformation is simply to re-scale the constant $\kappa$ as

$$\kappa \longrightarrow \frac{\kappa}{1 + \lambda^2 \kappa^2} . \tag{165}$$

Therefore the scalar Plebanski Lagrangian $\mathcal{L}_{sPl}$ is a fixed point of the $2d$ classical $T\overline{T}$ flow, in the sense that the $T\overline{T}$ deformation sends one theory in this class to another theory within the same class which has a different value of $\kappa$ and an additive constant in the Lagrangian, neither of which affects the equations of motion.

Exactly as in the gauge theory case, we can also obtain a modified form of the scalar Plebanski theory which satisfies a $\gamma$-flow equation driven by root-$T^2$. This theory is described by the Lagrangian

$$\mathcal{L}_{\gamma sPl} = \frac{\sqrt{2}\kappa}{\sqrt{x_2 - x_1^2}} \left( \cosh(\gamma) x_1 + \sinh(\gamma) \sqrt{2x_2 - x_1^2} \right) . \tag{166}$$

The modified scalar Plebanski theory satisfies the flow equation

$$\frac{\partial \mathcal{L}_{\gamma sPl}}{\partial \gamma} = R = \sqrt{\frac{1}{2} T^{\mu\nu} T_{\mu\nu} - \frac{1}{4} \left( T^\mu{}_\mu \right)^2} , \tag{167}$$

where $R$ is the usual root-$T^2$ combination. Furthermore, the stress tensor $T_{\mu\nu}$ associated with $\mathcal{L}_{\gamma sPl}$ has the properties

$$O_{T^2} = 2\kappa^2 , \qquad T^\mu{}_\mu = 2\mathcal{L}_{\gamma sPl} , \tag{168}$$

which means that it falls into the class of $T^2$ fixed point theories considered in Section 2.3. We can thus immediately write down the solution for the $T^2$ flow equation with the seed theory $\mathcal{L}_{\gamma sPl}$ at $\lambda = 0$, which is

$$\mathcal{L}_{(\lambda, \gamma)sPl} = \frac{\mathcal{L}_{\gamma sPl}}{1 + \kappa^2 \lambda^2} + \frac{2\lambda \kappa^2}{1 + \lambda^2 \kappa^2} . \tag{169}$$

The equations of motion for the modified scalar Plebanski theory $\mathcal{L}_{\gamma sPl}$, at any value of $\gamma$, are unchanged under the two-dimensional $T\overline{T}$ flow since the effect of the deformation is merely an overall rescaling of the Lagrangian and a shift by a constant. However, exactly as in the $4d$ case, the $\gamma$-deformation non-trivially modifies the model.

### Reverse modified-Nambu-Goto

There also exist scalar analogues of the reverse Born-Infeld theory and its ModMax-like extension. In fact, as in the gauge theory case, there is an entire $U(1)$'s worth of these theories parameterized by an angle $\theta$. First consider the family of Lagrangians

$$\mathcal{L}_{\theta NG} = \frac{\alpha e^{i\theta}}{\lambda} \left( 1 - \sqrt{1 - \lambda x_1 - \frac{\lambda^2}{2} \left( x_2 - x_1^2 \right)} \right) + \beta \sqrt{x_2 - x_1^2} . \tag{170}$$

Like its gauge theory analogue, the Lagrangian $\mathcal{L}_{\theta NG}$ satisfies the scalar zero-birefringence conditions (152) but not the scalar electric-magnetic duality condition (153). Furthermore, this family of Lagrangians satisfies the flow equation

$$\frac{\partial \mathcal{L}_{\theta NG}}{\partial \lambda} = \frac{e^{-i\theta}}{4\alpha} \left( T^{\mu\nu} T_{\mu\nu} - \left( T^\mu{}_\mu \right)^2 \right) , \tag{171}$$

for any values of $\theta, \alpha, \beta$. When $\theta = 0$ and $\alpha = 1$, this reduces to the usual two-dimensional flow equation which yields the Nambu-Goto Lagrangian for a static gauge string in three target spacetime dimensions. However, for $\theta = -\frac{\pi}{2}$, the Lagrangian becomes

$$\mathcal{L}_{\text{rNG}} = \frac{\alpha}{\lambda}\left(\sqrt{-1 + \lambda x_1 + \frac{\lambda^2}{2}\left(x_2 - x_1^2\right)} - i\right) + \beta \sqrt{x_2 - x_1^2}. \tag{172}$$

This is the scalar analogue of the reverse Born-Infeld Lagrangian (95), including the subtraction of an imaginary constant so that the theory has a finite limit as $\lambda \to 0$.

It is straightforward to write down a $\gamma$-flowed version of this family of Lagrangians,

$$\mathcal{L}_{(\gamma,\theta)\text{NG}} = \frac{\alpha e^{i\theta}}{\lambda}\left(1 - \sqrt{1 - \lambda\left(\cosh(\gamma)x_1 + \sinh(\gamma)\sqrt{2x_2 - x_1^2}\right) - \frac{\lambda^2}{2}\left(x_2 - x_1^2\right)}\right) + \beta\sqrt{x_2 - x_1^2}. \tag{173}$$

The Lagrangian (173) satisfies the same flow equation (171) at any value of $\gamma$, and in the limit as $\gamma \to 0$ this Lagrangian reduces to the expression $\mathcal{L}_{\theta\text{NG}}$ that we considered before. It also satisfies a second $\gamma$-flow driven by a marginal combination of stress tensors,

$$\frac{\partial \mathcal{L}_{(\gamma,\theta)\text{NG}}}{\partial \gamma} = \sqrt{\frac{1}{2}T^{\mu\nu}T_{\mu\nu} - \frac{1}{4}\left(T^{\mu}{}_{\mu}\right)^2}, \tag{174}$$

which is the appropriate root-$T^2$ operator for this class of theories. The whole family of generalized Nambu-Goto-type Lagrangians $\mathcal{L}_{(\gamma,\theta)\text{NG}}$ therefore satisfies two commuting flow equations, the irrelevant flow driven by the $T\overline{T}$ operator and the marginal flow driven by the root-$T\overline{T}$ operator, at any value of $\theta$.

Again, as in the gauge theory context, we can re-interpret the irrelevant flow by defining a complex flow parameter

$$\widehat{\lambda} = \frac{\lambda}{\alpha e^{i\theta}}, \tag{175}$$

so that the $T\overline{T}$ flow satisfied by this family of theories can be written as

$$\frac{\partial \mathcal{L}_{(\gamma,\theta)\text{NG}}}{\partial \widehat{\lambda}} = \frac{1}{4}\left(T^{\mu\nu}T_{\mu\nu} - \left(T^{\mu}{}_{\mu}\right)^2\right) = O_{T^2}. \tag{176}$$

We see that these theories can be formally viewed as arising from a $T\overline{T}$ deformation with a complex value of the flow parameter and the appropriate initial condition. All of the discussion following equation (103), which is the corresponding complex flow equation for the $4d$ version of this theory, also applies to the scalar setting. For instance, the existence of the complex flows can be interpreted as a consequence of the observation that the Lagrangian may be promoted to a holomorphic function of a complex variable $\lambda$.

Finally, we note that one can define a subtracted version of this family of Lagrangians,

$$\widetilde{\mathcal{L}}_{(\gamma,\theta)\text{NG}} = \mathcal{L}_{(\gamma,\theta)\text{NG}} - \frac{\alpha e^{i\theta}}{\lambda}, \tag{177}$$

for which the combination $O_{T^2}$ is a constant. Written in terms of the tension variable $T = \frac{1}{\lambda}$, this Lagrangian is

$$\widetilde{\mathcal{L}}_{(\gamma,\theta)\text{NG}} = -\alpha e^{i\theta}\sqrt{T^2 - T\left(\cosh(\gamma)x_1 + \sinh(\gamma)\sqrt{2x_2 - x_1^2}\right) - \frac{1}{2}\left(x_2 - x_1^2\right)} + \beta\sqrt{x_2 - x_1^2}. \tag{178}$$

Given the stress tensor $\widetilde{T}_{\mu\nu}$ associated with (178), the resulting $T\overline{T}$ operator is constant:

$$\widetilde{O}_{T^2} = \frac{1}{4}\left(\widetilde{T}^{\mu\nu}\widetilde{T}_{\mu\nu} - \left(\widetilde{T}^{\mu}{}_{\mu}\right)^2\right) = -\frac{1}{2}e^{2i\theta}\alpha^2 T^2. \tag{179}$$

As a consequence of the general analysis of Section 2.2, this theory satisfies

$$\frac{\partial \widetilde{\mathcal{L}}_{(\gamma,\theta)\text{NG}}}{\partial T} = \frac{\alpha \widetilde{T}^{\mu}{}_{\mu}}{\sqrt{2\left|\widetilde{T}^{\rho\sigma}\widetilde{T}_{\rho\sigma} - \left(\widetilde{T}^{\rho}{}_{\rho}\right)^2\right|}}, \tag{180}$$

which is another example of a $T\bar{T}$-like flow driven by a relevant operator.

# 5 Supersymmetric flows in 4$d$

In the previous sections we have described several $T\bar{T}$-like flows. In this section we aim at reporting on two results. Firstly, we will extend the results of [49] and prove that the supersymmetric 4$d$ $\mathcal{N}=1$ ModMax-Born-Infeld theory proposed in [46, 47] (see also [82] for $\mathcal{N}=1,2$ supersymmetric ModMax) not only satisfies a $\lambda$-flow, as shown in [15, 49], but also an appropriate supersymmetric $\gamma$-flow which extends the bosonic result of [48]. Since all the results in this section can be formally extended to complex values of $\lambda$, our results will apply to both Modified Born-Infeld and Reverse Born-Infeld theories up to adding appropriate imaginary terms in the full superspace Lagrangian. Secondly, we will demonstrate that a supersymmetric extension of the Modified Plebanski theory satisfies the same flow equations. We will finish the section by commenting on supersymmetric extensions of the $\sqrt{\hat{T}^{\mu\nu}\hat{T}_{\mu\nu}}$ operator which drives $\gamma$ deformations.

## 5.1 Review of the supersymmetric $\lambda$-flow for 4$d$ $\mathcal{N}=1$ ModMax-BI

The superspace Lagrangian for the 4$d$ $\mathcal{N}=1$ Modified Born-Infeld theory, also denoted $\gamma$BI, can be written in the following form [49]

$$\mathcal{L} = \frac{\cosh(\gamma)}{4}\left[\int d^2\theta\, W^2 + \int d^2\bar{\theta}\, \bar{W}^2 + \int d^2\theta d^2\bar{\theta}\, W^2\bar{W}^2 K(\mathbb{S},\mathbb{P})\right], \tag{181a}$$

where the function $K(\mathbb{S},\mathbb{P})$ for $\gamma$BI is given by

$$K_{\gamma\text{BI}}(\mathbb{S},\mathbb{P}) = \frac{T - \sqrt{T^2 - 2T\left[\cosh(\gamma)\mathbb{S} + \sinh(\gamma)\sqrt{\mathbb{S}^2 + \mathbb{P}^2}\right] - \mathbb{P}^2} - \cosh(\gamma)\mathbb{S}}{\cosh(\gamma)(\mathbb{S}^2 + \mathbb{P}^2)}, \tag{181b}$$

and the superfields $\mathbb{S}$ and $\mathbb{P}$ are defined as follows[13]

$$\mathbb{S} = -\frac{1}{2}(u + \bar{u}), \qquad \mathbb{P} = \frac{\text{i}}{2}(u - \bar{u}), \qquad u = \frac{1}{8}D^2 W^2, \qquad \bar{u} = \frac{1}{8}\bar{D}^2\bar{W}^2, \tag{182}$$

such that $\mathbb{S}^2 + \mathbb{P}^2 = u\bar{u}$. The expressions $W^2 = W^\alpha W_\alpha$ and $\bar{W}^2 = \bar{W}_{\dot{\alpha}}\bar{W}^{\dot{\alpha}}$ are defined in terms of the superfield strength of a 4$d$, $\mathcal{N}=1$ Abelian vector multiplet $W_\alpha$, and its conjugate $\bar{W}_{\dot{\alpha}} = (W_\alpha)^*$, satisfying

$$\bar{D}_{\dot{\beta}}W_\alpha = 0, \quad D^\alpha W_\alpha = \bar{D}_{\dot{\alpha}}\bar{W}^{\dot{\alpha}}, \tag{183}$$

which is equivalent to the following expansion in terms of the component fields describing the vector multiplet:

$$W_\alpha = -\text{i}\lambda_\alpha + \theta_\alpha \text{D} - \text{i}(\sigma^{\mu\nu}\theta)_\alpha F_{\mu\nu} + \theta^2(\sigma^\mu\partial_\mu\bar{\lambda})_\alpha. \tag{184}$$

---

[13]We refer the reader to [15, 49] for more details about our 4$d$ $\mathcal{N}=1$ superspace conventions. For example, we use the compact expression $D^2 = D^\alpha D_\alpha$, together with its complex conjugate $\bar{D}^2 = \bar{D}_{\dot{\alpha}}\bar{D}^{\dot{\alpha}}$, defined in terms of the superspace covariant spinor derivatives $D_\alpha$ and $\bar{D}^{\dot{\alpha}}$.

Here the complex spinor $\lambda_\alpha$ is the gaugino, D is the real auxiliary field, and $F_{\mu\nu} = 2\partial_{[\mu}\nu_{\nu]}$ is the field strength of an Abelian connection $\nu_\mu$. The superfields $\mathbb{S}$ and $\mathbb{P}$ are related to the scalar combinations of $F_{\mu\nu}$ used in the previous sections, $S$ and $P$ of eq. (11), through the following $\theta = 0$ reduction

$$\mathbb{S}|_{\theta=0} = S + \frac{1}{2}D^2, \qquad \mathbb{P}|_{\theta=0} = P. \tag{185}$$

It is useful to rewrite $K_{\gamma\mathrm{BI}}(\mathbb{S}, \mathbb{P})$ in terms of $u, \bar{u}$, and $\lambda = 1/T$:

$$K_{\gamma\mathrm{BI}}(u, \bar{u}) = \frac{1 - \sqrt{1 + \lambda\left[\cosh(\gamma)(u+\bar{u}) - 2\sinh(\gamma)\sqrt{u\bar{u}}\right] + \frac{1}{4}\lambda^2(u-\bar{u})^2}}{\lambda\cosh(\gamma)u\bar{u}} + \frac{1}{2\lambda}\left(\frac{1}{u} + \frac{1}{\bar{u}}\right), \tag{186}$$

The supersymmetric ModMax theory is then obtained by taking the limit $\lambda \to 0$, while setting $\gamma = 0$ leads to the supersymmetric Maxwell-Born-Infeld Lagrangian proposed by Bagger and Galperin in [83].

In [15, 49] it was shown that (181) with the choice of $K_{\gamma\mathrm{BI}}$ given above satisfies the following supercurrent-squared flow[14]

$$\frac{\partial \mathcal{L}_{\mathrm{susy}-\gamma\mathrm{BI}}}{\partial \lambda} = \frac{1}{8}\int d^2\theta d^2\bar{\theta}\, \mathcal{O}_{T^2}, \qquad \mathcal{O}_{T^2} = \frac{1}{16}\mathcal{J}^{\alpha\dot{\alpha}}\mathcal{J}_{\alpha\dot{\alpha}} - \frac{5}{8}\mathcal{X}\bar{\mathcal{X}}, \tag{187}$$

where the superfields $\mathcal{J}_{\alpha\dot{\alpha}}$ and $\mathcal{X}$ define the Ferrara-Zumino (FZ) supercurrent multiplet [84] satisfying

$$D^\alpha \mathcal{J}_{\alpha\dot{\alpha}} = \bar{D}_{\dot{\alpha}}\bar{\mathcal{X}}, \qquad \bar{D}_{\dot{\alpha}}\mathcal{X} = 0. \tag{188}$$

The flow holds once some implications of the equations of motion are used – see discussions in [15, 49] and in the next subsection. Note that the relative coefficient between $\mathcal{J}^{\alpha\dot{\alpha}}\mathcal{J}_{\alpha\dot{\alpha}}$ and $\mathcal{X}\bar{\mathcal{X}}$ is uniquely fixed by requiring the operator $\frac{1}{8}\mathcal{O}_{T^2}$ to describe a supersymmetric extension of the bosonic operator $O_{T^2}$ in eq. (27) for $d = 4$, see [15] for details. For the model (181) it can be shown to hold

$$\frac{1}{8}\int d^2\theta d^2\bar{\theta}\, \mathcal{O}_{T^2} = \frac{1}{8}\left(T^{\mu\nu}T_{\mu\nu} - \frac{1}{2}(T^\mu{}_\mu)^2\right) + \text{fermions} = O_{T^2} + \text{fermions}. \tag{189}$$

## 5.2 Supersymmetric $\gamma$-flow for 4d $\mathcal{N} = 1$ ModMax-BI

A very similar calculation to the one of [15, 49] shows that the Lagrangian (181) satisfies the following $\gamma$-flow equation:

$$\frac{\partial \mathcal{L}_{\mathrm{susy}-\gamma\mathrm{BI}}}{\partial \gamma} = \frac{1}{2}\int d^2\theta d^2\bar{\theta}\, \mathcal{R}_\gamma, \tag{190a}$$

where the root superspace operator $\mathcal{R}$ is given by

$$\mathcal{R} := \frac{a\mathcal{J}^{\alpha\dot{\alpha}}\mathcal{J}_{\alpha\dot{\alpha}} + b\bar{\mathcal{X}}\mathcal{X}}{\sqrt{([D^{(\gamma}, \bar{D}_{(\dot{\gamma}}]\mathcal{J}^{\delta)}{}_{\dot{\delta})})[D_{(\gamma}, \bar{D}^{(\dot{\gamma}}]\mathcal{J}_{\delta)}{}^{\dot{\delta})}}}, \quad a = -b = 1. \tag{190b}$$

We have left coefficients $a$ and $b$ in the numerator, which should then be set to $a = -b = 1$, for convenience of the following discussions. Note that the subscript $\gamma$ in (190a) indicates that

---

[14]Note that in [15, 49] we used $\alpha^2 = \lambda$ to parametrise the inverse of the brane tension $T$.

the superspace operator $\mathcal{R}$ defined above is evaluated for the theory with value $\gamma$ along the flow. For the supersymmetric $\gamma$BI model the operator $\mathcal{R}$ can be proven to satisfy

$$\frac{1}{2}\int d^2\theta d^2\bar{\theta}\,\mathcal{R} = R + \text{fermions} = \frac{1}{2}\sqrt{\hat{T}^{\mu\nu}\hat{T}_{\mu\nu}} + \text{fermions}, \tag{191}$$

following the normalization of $R$ given in (110), as expected for a supersymmetric extension of the $\gamma$BI flow. Let us now turn to proving the previous statements.

A key technical step in the computation of supersymmetric $T\bar{T}$-like flows is the construction of the supercurrent multiplet. For the Lagrangian (181), the Ferrara-Zumino multiplet was derived in [15,49] by using the results of [85]. The superfields $\mathcal{J}_{\alpha\dot{\alpha}}$ and $\mathcal{X}$ can be written as:

$$\mathcal{X} = \frac{4\cosh(\gamma)}{3}W^2\bar{u}\left(\Gamma + \bar{\Gamma} - K\right) + W^2\bar{W}(\cdots), \tag{192a}$$

$$\mathcal{J}_{\alpha\dot{\alpha}} = \cosh(\gamma)\left\{-4W_\alpha\bar{W}_{\dot{\alpha}}\left(1 - \bar{u}\Gamma - u\bar{\Gamma}\right) + \frac{1}{6}(D_\alpha W^2)(\bar{D}_{\dot{\alpha}}\bar{W}^2)\left(\Gamma + \bar{\Gamma} - K\right)\right\}$$
$$+ W^2\bar{W}(\cdots) + \bar{W}^2 W(\cdots), \tag{192b}$$

where the ellipsis are contributions that vanish identically when evaluating $\mathcal{R}_\gamma$ due to the nilpotency conditions $W_\alpha W_\beta W_\gamma = 0$ and $\bar{W}^{\dot{\alpha}}\bar{W}^{\dot{\beta}}\bar{W}^{\dot{\gamma}} = 0$. The superfields $\Gamma = \Gamma(u,\bar{u})$, and $\bar{\Gamma} = \bar{\Gamma}(u,\bar{u})$ are defined as follows

$$\Gamma(u,\bar{u}) = \frac{\partial\left(uK(u,\bar{u})\right)}{\partial u}, \qquad \bar{\Gamma}(u,\bar{u}) = \frac{\partial\left(\bar{u}K(u,\bar{u})\right)}{\partial\bar{u}}. \tag{193}$$

By using the previous expressions, it is straightforward to calculate $\mathcal{J}^{\alpha\dot{\alpha}}\mathcal{J}_{\alpha\dot{\alpha}}$ and $\bar{\mathcal{X}}\mathcal{X}$:

$$\mathcal{J}^{\alpha\dot{\alpha}}\mathcal{J}_{\alpha\dot{\alpha}} = 16\cosh^2(\gamma)W^2\bar{W}^2\left\{\left(1 - \bar{u}\Gamma - u\bar{\Gamma}\right)^2 + \frac{1}{9}u\bar{u}\left(\Gamma + \bar{\Gamma} - K\right)^2\right\}$$
$$- \frac{4\cosh^2(\gamma)}{3}W^2\bar{W}^2(D^\alpha W_\alpha)^2\left(1 - \bar{u}\Gamma - u\bar{\Gamma}\right)\left(\Gamma + \bar{\Gamma} - K\right), \tag{194a}$$

$$\mathcal{X}\bar{\mathcal{X}} = \frac{16\cosh^2(\gamma)}{9}W^2\bar{W}^2 u\bar{u}\left(\Gamma + \bar{\Gamma} - K\right)^2. \tag{194b}$$

The second line of (194a) is further simplified if we use the following equation

$$W^2\bar{W}^2(D^\alpha W_\alpha) = 0. \tag{195}$$

This was proven in Appendix A of [15], see also the discussion in [49]. It turns out that this condition is an implication of the equations of motions of $W_\alpha$ and hold for any model of the form (181) with any function $K$, and related $\Gamma$ and $\bar{\Gamma}$.[15] Importantly, the condition (195) can be understood as the fact that the auxiliary field of the vector multiplet satisfies $\mathsf{D} \propto D^\alpha W_\alpha|_{\theta=0} = 0 + \text{fermions}$, which is equivalent to having $\mathcal{N} = 1$ supersymmetry preserved on-shell.

The need to use this condition when analysing the flows of our interest can be highlighted by looking at the bosonic truncation of the super ModMax component Lagrangian

$$\mathcal{L}_{MM}^{(\text{bos})} = \cosh(\gamma)\left(S + \frac{1}{2}\mathsf{D}^2\right) + \sinh(\gamma)\sqrt{\left(S + \frac{1}{2}\mathsf{D}^2\right)^2 + P^2}. \tag{196}$$

---

[15]These models include the class of supersymmetric theories satisfying electric-magnetic duality conditions identified in [76,86].

One can ask whether this obeys a regular (non-supersymmetrised) $\sqrt{T\bar{T}}$ flow driven by $R$. The $\sqrt{T\bar{T}}$ operator of the above model is given by

$$R = \sinh(\gamma)\left(S + \frac{1}{2}\mathrm{D}^2\right) + \cosh(\gamma)\frac{(S^2 + P^2)}{\sqrt{\left(S + \frac{1}{2}\mathrm{D}^2\right)^2 + P^2}}\,. \tag{197}$$

Comparing this to the derivative with respect to $\gamma$

$$\frac{\partial \mathcal{L}_{MM}^{(\text{bos})}}{\partial \gamma} = \sinh(\gamma)\left(S + \frac{1}{2}\mathrm{D}^2\right) + \cosh(\gamma)\sqrt{\left(S + \frac{1}{2}\mathrm{D}^2\right)^2 + P^2}\,, \tag{198}$$

it is easy to see that the flow equation will only be satisfied when the auxiliary field $\mathrm{D} = 0$, however, without imposing any other equation of motion. It can in fact be proven that this is true for the entire multiplet of fields described by $W_\alpha$ and that the constraint (195) is only imposing an equation of motion for the auxiliary field D, but not for the gaugino $\lambda_\alpha$ nor the Maxwell field strength $F_{\mu\nu}$. The same argument extends to the $\gamma$BI case.

Coming back to the proof of the supersymmetric flows, we note that for the $\gamma$BI theory, it can be shown that (195) is satisfied – see [47, 49]. As a final step in our derivation, it is necessary to evaluate the denominator in (190b). Thanks to nilpotency conditions and eq. (195), it holds

$$W^2\bar{W}^2\big([D^{(\gamma},\bar{D}_{(\dot\gamma}]\mathcal{J}^{\delta)}{}_{\dot\delta)})[D_{(\gamma},\bar{D}^{(\dot\gamma}]\mathcal{J}_{\delta)}{}^{\dot\delta)} = 32^2 W^2\bar{W}^2 u\bar{u}\big(1 - \bar{u}\Gamma - u\bar{\Gamma}\big)^2\,, \tag{199a}$$

$$W^2\bar{W}^2\Big[\big([D^{(\gamma},\bar{D}_{(\dot\gamma}]\mathcal{J}^{\delta)}{}_{\dot\delta)})[D_{(\gamma},\bar{D}^{(\dot\gamma}]\mathcal{J}_{\delta)}{}^{\dot\delta)}\Big]^{-\frac{1}{2}} = \frac{W^2\bar{W}^2}{32\sqrt{u\bar{u}}\big(1 - \bar{u}\Gamma - u\bar{\Gamma}\big)}\,, \tag{199b}$$

where we have chosen a positive root close to the identity. By using this result, the superfield $\mathcal{R}$ in (190b) simplifies to

$$\mathcal{R} = \cosh(\gamma)W^2\bar{W}^2\big(1 - \bar{u}\Gamma - u\bar{\Gamma}\big)\left\{\frac{a}{2\sqrt{u\bar{u}}} + \frac{(a+b)\sqrt{u\bar{u}}}{18}\frac{\big(\Gamma + \bar{\Gamma} - K\big)^2}{\big(1 - \bar{u}\Gamma - u\bar{\Gamma}\big)^2}\right\}\,. \tag{200}$$

For the flow (190a) to hold it is necessary to choose $a = 1$ and $b = -1$. Note that if $K = \Gamma + \bar{\Gamma}$ the theory is superconformal, as for example the supersymmetric ModMax theory, and, due to the fact that in this case $\mathcal{X} = \bar{\mathcal{X}} = 0$, the second term in (200) is identically zero independently of the choice $b = -1$. When one chooses $a = -b = 1$ in the operator (190b), the expression for $\mathcal{R}_\gamma$ computed for the Lagrangian (181) and associated function $K(u,\bar{u})$, eq. (186), takes the simplified form:

$$\mathcal{R}_\gamma = \frac{2\sqrt{u\bar{u}}\cosh(\gamma) - \sinh(\gamma)(u + \bar{u})}{4u\bar{u}\sqrt{1 + \lambda\cosh(\gamma)(u + \bar{u}) - 2\lambda\sinh(\gamma)\sqrt{u\bar{u}} + \frac{1}{4}\lambda^2(u - \bar{u})^2}}\,. \tag{201}$$

As wanted, this is precisely twice the left-hand side of eq. (190a).

To conclude, let us show that equation (191) is satisfied. The Ferrara-Zumino supercurrent multiplet comprises the following component fields within the superfields $\mathcal{J}_{\alpha\dot\alpha}$ and $\mathcal{X}$: a vector $j_\mu(x)$, the complex conserved spinor current $S_\mu(x)$, a complex scalar field $\mathsf{x}(x)$, and the conserved stress tensor $T_{\mu\nu}$. We refer the reader to [15,49] for details and the results concerning the Ferrara-Zumino multiplet in our notation. For the purposes of our paper it suffices to describe the dependence of $\mathcal{J}_{\alpha\dot\alpha}$ and $\mathcal{X}$ upon the stress tensor $T_{\mu\nu}$ only. It holds

$$\mathcal{J}_{\alpha\dot\alpha}(x,\theta,\bar\theta) = -4\theta^\beta\bar\theta^{\dot\beta}(\sigma^\mu)_{\alpha\dot\alpha}(\sigma^\nu)_{\beta\dot\beta}T_{\mu\nu} - \frac{8}{3}\theta_\alpha\bar\theta_{\dot\alpha}\Theta + \cdots\,, \tag{202a}$$

$$\mathcal{X}(x,\theta) = \frac{2}{3}\theta^2\Theta(x) + \cdots\,, \qquad \Theta(x) := T^\mu{}_\mu(x)\,. \tag{202b}$$

Note that in the case of the vector multiplet models we are interested in, the bosonic component operators in the Ferrara-Zumino multiplet, $j_\mu$ and x, are purely fermionic and at least quadratic in the gauginos $\lambda_\alpha := iW_\alpha|_{\theta=0}$ and $\bar\lambda^{\dot\alpha} = -i\bar W^{\dot\alpha}|_{\theta=0}$. This can be easily seen by noticing that $j_\mu = \mathcal{J}_\mu|_{\theta=0}$ and by looking at the explicit form of the supercurrents in (192). Hence, the ellipsis in (202) are all functions of fermionic component fields. By using eq. (202) one obtains

$$\mathcal{J}^{\alpha\dot\alpha}\mathcal{J}_{\alpha\dot\alpha} = 16\theta^2\bar\theta^2\Big(T^{\mu\nu}T_{\mu\nu} - \frac{2}{9}\Theta^2\Big) + \cdots, \tag{203a}$$

$$\mathcal{X}\bar{\mathcal{X}} = \frac{4}{9}\theta^2\bar\theta^2\Theta^2 + \cdots \tag{203b}$$

Taking a linear combination of the previous expressions, it holds

$$a\mathcal{J}^{\alpha\dot\alpha}\mathcal{J}_{\alpha\dot\alpha} + b\mathcal{X}\bar{\mathcal{X}} = \theta^2\bar\theta^2\Big(16aT^{\mu\nu}T_{\mu\nu} + \frac{4b-32a}{9}\Theta^2\Big) + \cdots \tag{204}$$

Since the previous result is a $\theta^2\bar\theta^2$ term, in evaluating the denominator in (190b) we can consider the $\theta = 0$ term only. Note that, up to fermions, the combination $[D_{(\alpha},\bar D^{(\dot\alpha}]\mathcal{J}_{\beta)}{}^{\dot\beta)}|_{\theta=0}$ is proportional to $(\sigma^\mu)_{(\alpha}{}^{(\dot\alpha}(\sigma^\nu)_{\beta)}{}^{\dot\beta)}\hat T_{\mu\nu}$ with $\hat T_{\mu\nu} = T_{\mu\nu} - \frac{1}{4}g_{\mu\nu}\Theta$ being the traceless part of the stress tensor. This implies the following relation

$$\sqrt{([D^{(\gamma},\bar D_{(\dot\gamma}]\mathcal{J}^{\delta)}{}_{\dot\delta)})[D_{(\gamma},\bar D^{(\dot\gamma}]\mathcal{J}_{\delta)}{}^{\dot\delta)}}\Big|_{\theta=0} = \sqrt{\hat T^{\mu\nu}\hat T_{\mu\nu}} + \cdots \tag{205}$$

Written in terms of $\hat T_{\mu\nu}$ and $\Theta$, the operator (190b) satisfies

$$\int d^2\theta d^2\bar\theta\, \mathcal{R} = \Big(a\hat T^{\mu\nu}\hat T_{\mu\nu} + \frac{a+b}{36}\Theta^2\Big)\big(\hat T^{\mu\nu}\hat T_{\mu\nu}\big)^{-\frac{1}{2}} + \text{fermions}. \tag{206}$$

It is then clear that, as stated before, the only choice of coefficients $a$ and $b$ to obtain (191) from a superspace operator of the type (190b) is $a = -b = 1$.

To conclude this subsection we comment on the reversed $\gamma$BI model described by the bosonic Lagrangian (141). It is straightforward to show that a $\mathcal{N} = 1$ supersymmetric extension of (141) is given by

$$\mathcal{L}^{(\lambda,\gamma)}_{\text{susy-rBI}} = \int d^2\theta d^2\bar\theta\, \frac{16W^2\bar W^2}{(D^2W^2)(\bar D^2\bar W^2)}L^{(\lambda,\gamma)}_{\text{rBI}}(\mathbb{S},\mathbb{P}), \tag{207a}$$

$$L^{(\lambda,\gamma)}_{\text{rBI}}(\mathbb{S},\mathbb{P}) = \frac{\alpha}{\lambda}\sqrt{-1 + 2\lambda\Big(\cosh(\gamma)\mathbb{S} + \sinh(\gamma)\sqrt{\mathbb{S}^2 + \mathbb{P}^2}\Big) + \lambda^2\mathbb{P}^2} - \frac{\alpha i}{\lambda} + \beta\mathbb{P}. \tag{207b}$$

By analytic continuation of the analysis in this subsection, or by direct investigation, one can see that the previous Lagrangian satisfies the flow equations as (187) and (190a):

$$\frac{\partial\mathcal{L}^{(\lambda,\gamma)}_{\text{susy-rBI}}}{\partial\lambda} = \frac{i}{8\alpha}\int d^2\theta d^2\bar\theta\, \mathcal{O}_{T^2}, \qquad \mathcal{O}_{T^2} = \frac{1}{16}\mathcal{J}^{\alpha\dot\alpha}\mathcal{J}_{\alpha\dot\alpha} - \frac{5}{8}\mathcal{X}\bar{\mathcal{X}}, \tag{208a}$$

$$\frac{\partial\mathcal{L}^{(\lambda,\gamma)}_{\text{susy-rBI}}}{\partial\gamma} = \frac{1}{2}\int d^2\theta d^2\bar\theta\, \mathcal{R}, \qquad \mathcal{R} := \frac{\mathcal{J}^{\alpha\dot\alpha}\mathcal{J}_{\alpha\dot\alpha} - \bar{\mathcal{X}}\mathcal{X}}{\sqrt{([D^{(\gamma},\bar D_{(\dot\gamma}]\mathcal{J}^{\delta)}{}_{\dot\delta)})[D_{(\gamma},\bar D^{(\dot\gamma}]\mathcal{J}_{\delta)}{}^{\dot\delta)}}}. \tag{208b}$$

## 5.3 4d $\mathcal{N} = 1$ supersymmetric Plebanski theory and its flows

In this subsection we consider the supersymmetrisation of the $\lambda$ and $\gamma$ deformed Plebanski model described by the Lagrangians in equations (140) and (138). A straightforward supersymmetrisation of this model is achieved by considering the full superspace action

$$S^{(\lambda,\gamma)}_{\text{susy-Pl}} = \frac{\kappa}{1+\lambda^2\kappa^2}\int d^4x d^4\theta\, \frac{16W^2\bar W^2}{(D^2W^2)(\bar D^2\bar W^2)}\left[\frac{\cosh(\gamma)\mathbb{S} + \sinh(\gamma)\sqrt{\mathbb{S}^2+\mathbb{P}^2}}{\mathbb{P}} + \lambda\kappa\right]. \tag{209}$$

It is useful to rewrite the action in the form given by the Lagrangian (181a) with a specific function $K$. Up to total derivatives, the action (209) leads to the following

$$K_{\text{susy-Pl}}^{(\lambda,\gamma)}(u,\bar{u}) = \frac{1}{2u} + \frac{1}{2\bar{u}} + \frac{\kappa}{(1+\lambda^2\kappa^2)u\bar{u}} \left[ \frac{i(u+\bar{u}) + 2i\tanh(\gamma)\sqrt{u\bar{u}}}{u-\bar{u}} + \frac{\lambda\kappa}{\cosh(\gamma)} \right]. \qquad (210)$$

For a generic model described by the Lagrangian (181a) with an arbitrary function $K(u,\bar{u})$, when $W^2\bar{W}^2(D^\alpha W_\alpha) = 0$ is imposed, the $\mathcal{O}_{T^2}$ superfield of eq. (187) takes the form [49]

$$\mathcal{O}_{T^2} = \cosh^2(\gamma)W^2\bar{W}^2\left[ \left(1 - \bar{u}\Gamma - u\bar{\Gamma}\right)^2 - u\bar{u}\left(\Gamma + \bar{\Gamma} - K\right)^2 \right], \qquad (211)$$

where remember that $\Gamma$ and $\bar{\Gamma}$ are defined in (193). The superfield $\mathcal{R}$ takes in general the form

$$\mathcal{R} = \cosh(\gamma)W^2\bar{W}^2 \frac{1 - \bar{u}\Gamma - u\bar{\Gamma}}{2\sqrt{u\bar{u}}}, \qquad (212)$$

which is simply eq. (200) with $a = -b = 1$. Explicitly calculating (211) and (212) for the function $K_{\text{susy-Pl}}^{(\lambda,\gamma)}(u,\bar{u})$ of eq. (210), one finds:

$$\mathcal{O}_{T^2} = -i\kappa^2 \frac{2\kappa\lambda\cosh(\gamma)(u+\bar{u}) - 4\kappa\lambda\sinh(\gamma)\sqrt{u\bar{u}} + i(1-\kappa^2\lambda^2)(u-\bar{u})}{(1+\kappa^2\lambda^2)u\bar{u}(u-\bar{u})}, \qquad (213a)$$

$$\mathcal{R} = i\kappa \frac{\sinh(\gamma)(u+\bar{u}) - 2\cosh(\gamma)\sqrt{u\bar{u}}}{2(u-\bar{u})(1+\lambda^2\kappa^2)u\bar{u}}. \qquad (213b)$$

Comparing these to the derivatives of (209) with respect to $\lambda$ and $\gamma$, it can be shown that the following two flow equations hold:

$$\frac{\partial S_{\text{susy-Pl}}^{(\lambda,\gamma)}}{\partial\lambda} = \frac{1}{4}\int d^4x d^4\theta\, \mathcal{O}_{T^2}, \qquad \frac{\partial S_{\text{susy-Pl}}^{(\lambda,\gamma)}}{\partial\gamma} = \frac{1}{2}\int d^4x d^4\theta\, \mathcal{R}. \qquad (214)$$

As in the bosonic case, it is evident that the supersymmetric $\lambda$-deformation rescales the action and shifts the Lagrangian by a $\kappa$ dependent term. It is however interesting to note that in the supersymmetric case the shift described by the last term in (209) not only adds a constant term in the action but it also introduces new purely fermionic terms in the bosonic action.

## 5.4 On supersymmetric extensions of $\sqrt{\hat{T}^{\mu\nu}\hat{T}_{\mu\nu}}$

The analysis given in subsection 5.2 makes it clear that the superfield $\mathcal{R}$ in eq. (190b) is a supersymmetric extension of $R$. The reader might have wondered if such a supersymmetric extension is unique. It turns out that allowing for non-analyticity in stress-tensor operators makes it possible to construct several different supersymmetric extensions of $R$. We do not attempt a complete classification of such extensions here, but we will give some examples and simple arguments before concluding this section. For simplicity, we restrict our discussion to describing (classical) composite operators based on superconformal theories where we impose $\Theta \equiv 0$ and $\mathcal{X} = \bar{\mathcal{X}} \equiv 0$.

The logic in constructing the operator (190a) as a supersymmetric extension of $\sqrt{\hat{T}^{\mu\nu}\hat{T}_{\mu\nu}}$ was simple. We first identified a combination of descendants of $\mathcal{J}_{\alpha\dot{\alpha}}$ that includes $\hat{T}_{\mu\nu}$. As mentioned in subsection 5.2 this is precisely $[D_{(\alpha}, \bar{D}^{(\dot{\alpha}}]\mathcal{J}_{\beta)}{}^{\dot{\beta})}$. Then we constructed a superspace operator that includes $\sqrt{\hat{T}^{\mu\nu}\hat{T}_{\mu\nu}}$ among its $\theta = 0$ components, see eq. (205). Finally, we have engineered a simple fraction of two superfields constructed out of the supercurrent multiplet whose full superspace integral leads precisely to $\sqrt{\hat{T}^{\mu\nu}\hat{T}_{\mu\nu}}$, plus other possible terms that we

have not analysed in detail and we know that, at least for the models of the form (181a) with a generic function $K(u, \bar{u})$, are purely fermionic.

It is simple to show that other options would lead to alternative supersymmetric extensions of $\sqrt{\hat{T}^{\mu\nu}\hat{T}_{\mu\nu}}$. In principle one could write down operators of the form

$$\tilde{\mathcal{R}} = \mathcal{O}_{\text{higher-order}} \times \left[ ([D_{(\gamma}, \bar{D}^{(\dot{\gamma}}] \mathcal{J}_{\delta)}{}^{\dot{\delta})})[D^{(\gamma}, \bar{D}_{(\dot{\gamma}}] \mathcal{J}^{\delta)}{}_{\dot{\delta})}] \right]^{-c}, \tag{215}$$

for some constant exponent $c$. A necessary condition for consistency of the previous ansatz is that it holds

$$[\tilde{\mathcal{R}}] = [\mathcal{O}_{\text{higher-order}}] - 8c = 2 \iff [\mathcal{O}_{\text{higher-order}}] = 2 + 8c, \tag{216}$$

where $[X]$ denotes the mass dimension of $X$. As an example, for the operator in equation (190b), $c = \frac{1}{2}$ and $[\mathcal{J}^{\alpha\dot{\alpha}}\mathcal{J}_{\alpha\dot{\alpha}}] = [\bar{\mathcal{X}}\mathcal{X}] = 6$. A sufficient condition for $\tilde{\mathcal{R}}$ to give a supersymmetric extension of $\sqrt{\hat{T}^{\mu\nu}\hat{T}_{\mu\nu}}$ is that

$$\mathcal{O}_{\text{higher-order}} \propto \theta^2 \bar{\theta}^2 \left(\hat{T}^{\mu\nu}\hat{T}_{\mu\nu}\right)^{\frac{1}{2}+c} + \cdots \tag{217}$$

Let us search for operators satisfying these conditions.

If we neglect all component fields in $\mathcal{J}_{\alpha\dot{\alpha}}$ except the stress tensor, impose $\Theta = 0$, and also neglect vector derivatives of the stress tensor ($\partial_\mu T_{\nu\rho}$), the structure of the supercurrent and its descendants is very simple:

$$\mathcal{J}_{\alpha\dot{\alpha}}(x, \theta, \bar{\theta}) = -4\theta^\beta \bar{\theta}^{\dot{\beta}}(\sigma^\mu)_{\alpha\dot{\alpha}}(\sigma^\nu)_{\beta\dot{\beta}} T_{\mu\nu} + \cdots, \tag{218a}$$

$$\mathcal{J}_{\alpha\beta\dot{\beta}}(x, \theta, \bar{\theta}) := D_\alpha \mathcal{J}_{\beta\dot{\beta}}(x, \theta, \bar{\theta}) = -4\bar{\theta}^{\dot{\gamma}}(\sigma^\mu)_{\beta\dot{\beta}}(\sigma^\nu)_{\alpha\dot{\gamma}} T_{\mu\nu} + \cdots, \tag{218b}$$

$$\mathcal{J}_{\alpha\dot{\alpha}\dot{\beta}}(x, \theta, \bar{\theta}) := \bar{D}_{\dot{\alpha}} \mathcal{J}_{\beta\dot{\beta}}(x, \theta, \bar{\theta}) = -4\theta^\gamma (\sigma^\mu)_{\beta\dot{\beta}}(\sigma^\nu)_{\gamma\dot{\alpha}} T_{\mu\nu} + \cdots, \tag{218c}$$

$$\mathcal{J}_{\alpha\beta\dot{\alpha}\dot{\beta}}(x, \theta, \bar{\theta}) := [D_\alpha, \bar{D}_{\dot{\alpha}}]\mathcal{J}_{\beta\dot{\beta}}(x, \theta, \bar{\theta}) = -8(\sigma^\mu)_{\alpha\dot{\alpha}}(\sigma^\nu)_{\beta\dot{\beta}} T_{\mu\nu} + \cdots \tag{218d}$$

Note that in the superconformal case, the following symmetry properties hold:

$$\mathcal{J}_{\alpha\beta\dot{\beta}}(x, \theta, \bar{\theta}) = \mathcal{J}_{(\alpha\beta)\dot{\beta}}(x, \theta, \bar{\theta}), \qquad \mathcal{J}_{\alpha\dot{\alpha}\dot{\beta}}(x, \theta, \bar{\theta}) = \mathcal{J}_{\alpha(\dot{\alpha}\dot{\beta})}(x, \theta, \bar{\theta}), \tag{219}$$

$$\mathcal{J}_{\alpha\beta\dot{\alpha}\dot{\beta}}(x, \theta, \bar{\theta}) = \mathcal{J}_{(\alpha\beta)(\dot{\alpha}\dot{\beta})}(x, \theta, \bar{\theta}), \qquad (\sigma^\mu)_\alpha{}^{\dot{\alpha}}(\sigma^\nu)_\beta{}^{\dot{\beta}} T_{\mu\nu} = (\sigma^\mu)_{(\alpha}{}^{\dot{\alpha}}(\sigma^\nu)_{\beta)}{}^{\dot{\beta})} T_{\mu\nu}. \tag{220}$$

At this stage, it is simple to observe that there exists a unique superfield $\mathcal{O}_{\text{higher-order}}$ quadratic in the supercurrents satisfying the conditions described above and in particular eq. (217) – see the numerator of (190b). At cubic order in the supercurrent and its descendants, the following two Lorentz invariant combinations

$$\mathcal{J}^{\alpha\beta\dot{\alpha}\dot{\beta}} \mathcal{J}_{\alpha\dot{\alpha}}\mathcal{J}_{\beta\dot{\beta}}, \qquad \mathcal{J}^{\alpha\beta\dot{\alpha}} \mathcal{J}_{\alpha\dot{\alpha}\dot{\beta}}\mathcal{J}_\beta{}^{\dot{\beta}}, \tag{221}$$

are the only possible candidates. However, by using (218), it is simple to show that both the previous superfields are proportional to $\theta^2 \bar{\theta}^2 (T_\mu{}^\nu T_\nu{}^\rho T_\rho{}^\mu) + \cdots$, hence they do not satisfy (217). The next option is to consider operators quartic in $\mathcal{J}_{\alpha\dot{\alpha}}$, or derivatives thereof. It is not difficult to identify quartic operators that can satisfy (217) with $c = 3/2$. For instance, consider the following Lorentz invariant candidates for $\mathcal{O}_{\text{higher-order}}$

$$\mathcal{J}^{\alpha\beta\dot{\alpha}\dot{\beta}} \mathcal{J}_{\alpha\beta\dot{\alpha}\dot{\gamma}}\mathcal{J}^{\gamma\dot{\gamma}} \mathcal{J}_{\gamma\dot{\beta}}, \quad \mathcal{J}^{\alpha\beta\dot{\alpha}\dot{\beta}} \mathcal{J}_{\alpha\gamma\dot{\alpha}\dot{\beta}}\mathcal{J}^{\gamma\dot{\gamma}} \mathcal{J}_{\beta\dot{\gamma}}, \quad \mathcal{J}^{\alpha\beta\dot{\alpha}\dot{\beta}} \mathcal{J}_{\alpha\gamma\dot{\alpha}\dot{\gamma}}\mathcal{J}^{\gamma\gamma} \mathcal{J}_{\beta\dot{\beta}}, \tag{222a}$$

$$\mathcal{J}^{\alpha\beta\dot{\alpha}} \mathcal{J}_{\alpha\beta\dot{\alpha}}\mathcal{J}^{\gamma\dot{\beta}\dot{\gamma}} \mathcal{J}_{\gamma\dot{\beta}\dot{\gamma}}, \quad \mathcal{J}^{\alpha\beta\dot{\alpha}} \mathcal{J}_{\alpha\gamma\dot{\beta}}\mathcal{J}^{\gamma\dot{\beta}\dot{\gamma}} \mathcal{J}_{\beta\dot{\alpha}\dot{\gamma}}, \tag{222b}$$

where we neglected the combination $\mathcal{J}^{\alpha\beta\dot\alpha\dot\beta}\mathcal{J}_{\alpha\beta\dot\alpha\dot\beta}\mathcal{J}^{\gamma\dot\gamma}\mathcal{J}_{\gamma\dot\gamma}$ since it would be equivalent to considering the operator (190b). It is simple to show that the first two combinations in (222a) and the first in (222b) are proportional to $\theta^2\bar\theta^2(T_\mu{}^\nu T_\nu{}^\mu)^2 + \cdots$ while the combinations $\mathcal{J}^{\alpha\beta\dot\alpha\dot\beta}\mathcal{J}_{\alpha\gamma\dot\alpha\dot\gamma}\mathcal{J}^{\gamma\dot\gamma}\mathcal{J}_{\beta\dot\beta}$ and $\mathcal{J}^{\alpha\beta\dot\alpha}\mathcal{J}_{\alpha\gamma\dot\beta}\mathcal{J}^{\gamma\dot\beta\dot\gamma}\mathcal{J}_{\beta\dot\alpha\dot\gamma}$ are both proportional to $\theta^2\bar\theta^2(T_\mu{}^\nu T_\nu{}^\rho T_\rho{}^\tau T_\tau{}^\mu) + \cdots$. This implies that we have found four superfields

$$\frac{\mathcal{J}^{\alpha\dot\alpha}\mathcal{J}_{\alpha\dot\alpha}}{\left(([D^{(\gamma},\bar D_{(\dot\gamma}]\mathcal{J}^{\delta)}{}_{\dot\delta)})[D_{(\gamma},\bar D^{(\dot\gamma)}]\mathcal{J}_{\delta)}{}^{\dot\delta)}\right)^{\frac{1}{2}}} \,, \qquad \frac{\mathcal{J}^{\alpha\beta\dot\alpha\dot\beta}\mathcal{J}_{\alpha\beta\dot\alpha\dot\gamma}\mathcal{J}^{\gamma\dot\gamma}\mathcal{J}_{\gamma\dot\beta}}{\left(([D^{(\gamma},\bar D_{(\dot\gamma}]\mathcal{J}^{\delta)}{}_{\dot\delta)})[D_{(\gamma},\bar D^{(\dot\gamma)}]\mathcal{J}_{\delta)}{}^{\dot\delta)}\right)^{\frac{3}{2}}} \,, \qquad (223a)$$

$$\frac{\mathcal{J}^{\alpha\beta\dot\alpha\dot\beta}\mathcal{J}_{\alpha\gamma\dot\alpha\dot\beta}\mathcal{J}^{\gamma\dot\gamma}\mathcal{J}_{\beta\dot\gamma}}{\left(([D^{(\gamma},\bar D_{(\dot\gamma}]\mathcal{J}^{\delta)}{}_{\dot\delta)})[D_{(\gamma},\bar D^{(\dot\gamma)}]\mathcal{J}_{\delta)}{}^{\dot\delta)}\right)^{\frac{3}{2}}} \,, \qquad \frac{\mathcal{J}^{\alpha\beta\dot\alpha}\mathcal{J}_{\alpha\beta\dot\alpha}\mathcal{J}^{\gamma\dot\beta\dot\gamma}\mathcal{J}_{\gamma\dot\beta\dot\gamma}}{\left(([D^{(\gamma},\bar D_{(\dot\gamma}]\mathcal{J}^{\delta)}{}_{\dot\delta)})[D_{(\gamma},\bar D^{(\dot\gamma)}]\mathcal{J}_{\delta)}{}^{\dot\delta)}\right)^{\frac{3}{2}}} \,, \qquad (223b)$$

such that, considering $\Theta = 0$, they lead to manifestly supersymmetric extensions of $\sqrt{\hat T^{\mu\nu}\hat T_{\mu\nu}}$. The analysis could continue with operators $\mathcal{O}_{\text{higher-order}}$ of order higher than four in the supercurrents and for the non-conformal case ($\mathcal{X} \neq 0$) but we will not discuss this here.

Before finishing this section, some comments are in order. In the superconformal case, we have obtained alternative supersymmetric extensions of $\sqrt{\hat T^{\mu\nu}\hat T_{\mu\nu}}$. It is natural to ask whether these could have been used instead of $\mathcal{R}$ of eq. (190b) to define the supersymmetric ModMax $\gamma$-flow. Interestingly, for the superconformal models of the type (181a) with $K = \Gamma + \bar\Gamma$, which includes supersymmetric ModMax, all the operators constructed with the superfields in (223), up to setting $W^2\bar W^2(D^\alpha W_\alpha) = 0$, lead to the same combination

$$\tilde{\mathcal{R}} \propto \cosh(\gamma)W^2\bar W^2 \frac{1-\bar u\Gamma - u\bar\Gamma}{2\sqrt{u\bar u}} \,, \qquad (224)$$

which is precisely the superfield $\mathcal{R}$ of equation (212). The same is true for superfields $\tilde{\mathcal{R}}$ constructed out of the combinations $\mathcal{J}^{\alpha\beta\dot\alpha\dot\beta}\mathcal{J}_{\alpha\gamma\dot\alpha\dot\gamma}\mathcal{J}^{\gamma\dot\gamma}\mathcal{J}_{\beta\dot\beta}$ and $\mathcal{J}^{\alpha\beta\dot\alpha}\mathcal{J}_{\alpha\gamma\dot\beta}\mathcal{J}^{\beta\dot\beta\dot\gamma}\mathcal{J}_{\gamma\dot\alpha\dot\gamma}$ in eq. (222). Even the following superfields

$$\frac{\mathcal{J}^{\alpha\beta\dot\alpha\dot\beta}\mathcal{J}_{\alpha\dot\alpha}\mathcal{J}_{\beta\dot\beta}}{\left[([D_{(\gamma},\bar D^{(\dot\gamma)}]\mathcal{J}_{\delta)}{}^{\dot\delta)})(D^{(\gamma},\bar D_{(\dot\gamma}]\mathcal{J}^{\delta)}{}_{\dot\delta)}\right]^{\frac{7}{4}}} \,, \qquad \frac{\mathcal{J}^{\alpha\beta\dot\alpha}\mathcal{J}_{\alpha\dot\alpha\dot\beta}\mathcal{J}_\beta{}^{\dot\beta}}{\left[([D_{(\gamma},\bar D^{(\dot\gamma)}]\mathcal{J}_{\delta)}{}^{\dot\delta)})(D^{(\gamma},\bar D_{(\dot\gamma}]\mathcal{J}^{\delta)}{}_{\dot\delta)}\right]^{\frac{7}{4}}} \,, \qquad (225)$$

that we discarded being associated with $(T_\mu{}^\nu T_\nu{}^\rho T_\rho{}^\mu)$, for these models are proportional to the combination in (212). This indicates that, at least for supersymmetric ModMax, the flow is somehow unique. This is reassuring, since, up to ambiguities associated with different off-shell formulations, $\mathcal{N} = 1$ supersymmetric ModMax is expected to be the unique duality invariant and superconformal extension of supersymmetric Maxwell theory [46,47]. It would be interesting to check if this remains true for flows associated with non-conformal models, such as $\gamma$BI. We leave this for future investigations.

# 6 Conclusion

In this work, we have continued to explore the connections between classical stress tensor flows – with and without supersymmetry – and theories of nonlinear electrodynamics in four spacetime dimensions (along with their scalar analogues in $d = 2$).

Among our main results are the observations that the $4d$ root-$T^2$ operator can be written in a manifestly supersymmetric form using supercurrents in $\mathcal{N} = 1$ superspace, and the fact that $T^2$ flows in $d = 4$ are compatible with zero-birefringence conditions. These facts give

even more evidence that these stress tensor deformations are especially nice in the sense that they appear to preserve special properties of their seed theories.

We have also pointed out examples of theories which appear to be fixed points under $T^2$ deformations, such as the theory of Plebanski electrodynamics. A related but surprising result is that any theory which results from a $T^2$ flow of a conformal field theory also gives rise to a subtracted theory for which the combination $O_{T^2}$ is a constant.

There remain many interesting open questions, some of which we outline below. We hope to return to these questions in future work. A deeper understanding of these issues may well provide new insights on deformations of field theories and on the space of QFTs more generally.

### *Operator analysis of constant-$T\overline{T}$ and $T\overline{T}$ fixed point theories*

In Sections 2 and 3, we have studied certain theories for which the classical combination defining the $T^2$ operator appears to be a constant, independent of fields – and in some cases, where the equations of motion of the model are invariant under the $T\overline{T}$-like flow. Although these are classical statements, it would be interesting to study the quantum properties of the $T\overline{T}$ operator in such theories, at least in two spacetime dimensions where the operator is well-defined quantum mechanically. For instance, one might ask whether the subtracted Nambu-Goto Lagrangian has the property that the point-splitting procedure which defines $T\overline{T}$ produces an operator which is proportional to the identity.

Even more striking is the scalar Plebanski theory (161) which appears to be classically invariant under the $T\overline{T}$ flow. The property of being a "$T\overline{T}$ fixed point" likely cannot persist quantum mechanically, since general arguments imply that several oberservables are modified in a universal way under a $T\overline{T}$-deformation. For instance, the finite-volume spectrum on a cylinder of radius $R$ obeys an inviscid Burgers' equation under $T\overline{T}$ flow:

$$\frac{\partial}{\partial \lambda} E_n(R, \lambda) = E_n(R, \lambda) \frac{\partial}{\partial R} E_n(R, \lambda) + \frac{P_n(R)^2}{R} \,. \tag{226}$$

Thus it seems that the scalar Plebanski theory cannot genuinely remain invariant under a quantum $T\overline{T}$ deformation. Nonetheless, it is intriguing to ask what – if anything – is special about such classical $T\overline{T}$ fixed points at the quantum level. One might hope that a theory which is a $T\overline{T}$ fixed point in *any* sense might play a role similar to that of CFTs, which are fixed points under the conventional renormalization group flow.

### *Connections between subtracted $T^2$ theories and gravity*

As we mentioned above, the constant term appearing in the subtracted flows of Section 2 would act as a cosmological constant in a theory with dynamical gravity. It would be interesting to explore whether there is any gravitational interpretation of these subtracted theories for which the classical $T^2$ combination is a constant. We note that, for $\lambda < 0$, our subtracted flow is very similar to the combined bad-sign $T\overline{T}$ plus positive cosmological constant deformation which was proposed in [51] and further studied in [52, 53]. For $\lambda > 0$, the subtracted flow corresponds to a good-sign $T\overline{T}$ deformation along with a *negative* cosmological constant. One might also ask whether this is related to the behavior of the $T\overline{T}$ deformation on a space with constant negative curvature, which has been studied in [87, 88] and may be well-behaved at least in $d = 2$.

### *Deformations of p-form electrodynamics in higher dimensions*

In this manuscript we have focused on stress tensor deformations of a two-form field strength $F_{\mu\nu}$ in four spacetime dimensions, and the analogue of a scalar field $\phi$ (with a one-form field strength $\partial_\mu \phi$) in two spacetime dimensions. It is intriguing to ask whether there are similar

connections between $T\overline{T}$-like flows and theories of $p$-form electrodynamics in more general dimension. For instance, one could ask whether a $p$-form analogue of the zero-birefringence conditions (53) is preserved by some $T^2$ deformation for theories of a 3-form $H_{\mu\nu\rho}$ in $d = 6$. Some progress on $T\overline{T}$-like flows for $p$-form field strengths in $2p$ spacetime dimensions, focusing on Lagrangians which depend on only two Lorentz invariants and working to second order in $\lambda$, has appeared in [89]. A related question is whether the six-dimensional ModMax-type theory of a chiral tensor presented in [45] satisfies some kind of stress tensor flow. It may turn out that there is a more natural formulation of such theories to address this type of question, such as the formalism developed in [90] for ordinary electrodynamics and extended to the $p$-form case in [91].

### Supersymmetry, $T\overline{T}$-like and root-$T\overline{T}$-like deformations

There are still several directions that need more investigation concerning supersymmetry, superconformal symmetry, and the various (classically) irrelevant, marginal, and relevant $T\overline{T}$-like deformations. Let us mention a few directly related to the results in our paper.

In Section 2 we have used a simple argument to derive a *Trace Flow Equation* for a large class of classical flows defined in terms of operators that are functionals of the stress tensor whose seed theory is conformal. By using this, it was possible to obtain operators that were constant along flows and subtracted Lagrangians that satisfy relevant $T\overline{T}$-like flows. It would be interesting to obtain analog results with supersymmetry. For example, in [92] for the $2d$ $\mathcal{N} = (0,2)$ case, a superspace trace flow equation was proposed to analyse (at first order in $\lambda$) correlation functions of $T\overline{T}$-deformed superconformal models. It would be interesting to extend this result to other amounts of supersymmetry and space-time dimensions and to see how to use it for other types of flows.

As we have already alluded to in Section 5, it would be interesting to understand the degree to which supersymmetric extensions of the root-$T\overline{T}$ operator are unique. For the cases of $4d$ superconformal gauge theories studied in this work, we have checked that the various possible supersymmetric extensions appear to all be equivalent to the operator $\mathcal{R}$ which we have used to define our deformation. However, this might simply be due to the simplicity of Abelian vector multiplet models and the on-shell condition used. It is not obvious at all that this remains true for more general models and theories which are supersymmetric but not conformal. More generally, one should understand the possible ambiguities which are introduced by considering non-analytic combinations of currents (and supercurrents) more carefully.

One question that certainly deserves more investigation is whether and how root-$T\overline{T}$ deformations preserve supersymmetry and superconformal symmetry in general. In $d = 2$, it is well-established that a supersymmetric model remains supersymmetric under $T\overline{T}$ flows [10–14]. This can be made manifest by using superspace techniques. It remains an open question whether the operator (110) for $d = 2$ preserves supersymmetry in general, or whether it deforms supersymmetry in a controlled way.

In this paper, we have made several proposals for root-$T\overline{T}$-like deformations in $4d$, $\mathcal{N} = 1$ superspace that manifestly preserve supersymmetry. Analog operators can be defined in $d = 2$. For the supersymmetric models we have considered here, classical superconformal symmetry was preserved by the flow. However, due to the non-analytic denominators and the use of descendant superfields, the superspace operators defined as functionals of the Ferrara-Zumino supercurrent in section 5 appear to be conformal but not necessarily superconformal primaries. Hence they might not preserve superconformal symmetry in general. Understanding this problem in $2d$, even with a low amount of supersymmetry, might shed light on general properties of non-analytic marginal deformations of superconformal field theories.

*Further properties of root-$T\overline{T}$*

The study of marginal root-$T\overline{T}$-like deformations is in its infancy. Perhaps the most pressing issue is to understand whether these deforming operators can be defined at the quantum level. This is closely related to understanding the quantization of ModMax-type theories. Rewriting the ModMax theory, or its 2$d$ scalar analogue, in an equivalent form similar to those introduced in [93] might make the theory more amenable to quantization, in the same way that rewriting the Nambu-Goto Lagrangian in Polyakov form facilitates quantization in string theory.

Given the vast literature on the $T\overline{T}$ deformation in $d = 2$, it will also be interesting to see how many other results on this operator have analogues for root-$T\overline{T}$. For instance, there are several proposals for understanding double-trace $T\overline{T}$ holographically, including via a cutoff AdS$_3$ spacetime [55], modified boundary conditions [94], and other approaches [95]. Analogous mixed boundary conditions for the root-$T\overline{T}$ deformation will appear in [96], but it would be intriguing to understand marginal stress tensor deformations in holography more deeply, including their effects on observables such as gravitational Wilson lines which have been studied in the context of $T\overline{T}$ [97]. As a final example, one can couple two CFTs in a universal way using sequential $T\overline{T}$ deformations [98]; one might wonder whether there exists a similar procedure to couple two CFTs using root-$T\overline{T}$.

# Acknowledgments

C.F. thanks the University of Queensland for hospitality during a visit which led to some of the results in this work.

**Funding information** C.F. is supported by U.S. Department of Energy grant DE-SC0009999 and by funds from the University of California. L.S. is supported by a postgraduate scholarship at the University of Queensland. The work of G.T.-M. is supported by the Australian Research Council (ARC) Future Fellowship FT180100353, and by the Capacity Building Package of the University of Queensland.

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
