# Peer review of "Stress Tensor Flows, Birefringence in Non-Linear Electrodynamics, and Supersymmetry"

_SciPost Physics, doi:SciPost Phys. 15, 198 (2023)_

## Round 2 · Referee Report · Anonymous (Referee 1) · 2023-7-24

Strengths

1) The obtained results are novel and of considerable interest for a wide theoretical community. 2) The Conclusion contains a number of interesting and important open issues whose solution will contribute to the further development of this novel subject. 3) The paper is well written and is easy to follow.

Weaknesses

There is no real weaknesses but a couple of points need to be clarified (see report)

Report

The paper is dedicated to the review and study of various aspects of TTbar-like deformations of classical Abelian gauge field theories in four-dimensional space-time and corresponding scalar field theories in two space-time dimensions. The obtained results are novel and of considerable interest for a wide theoretical community. The results include the proof that the form of the stress-energy squared operator which drives,in particular, the TTbar-like deformation of Maxwell's electrodynamics to the Born-Infeld (BI) theory is the only one (modulo the trace of the stress-energy tensor) that preserve the property of no-birefringence of the non-linear electrodynamics. This was explicitly demonstrated on examples of the no-birefringence electrodynamics (BI, reverse-BI and Plebanski theory). This property singles out the form of the T^2 operator (2.20) in d=4 from other choices, at the classical level. (The corresponding choice in d=2 was previously known to define a local operator in a 2D quantum field theory). Other results are the identification of theories which are fixed points under TTbar-like deformations (such as Plebanski's electrodynamics) and the demonstration that to each theory which is a TTbar-like deformation of a conformal field theory there can be associated a theory with the subtracted (or added) "cosmological" constant in the Lagrangian for which the corresponding T^2 operator is constant. The paper also considers in detail properties of newly found marginal, so-called root-TTbar, deformed theories related to a recently discovered unique conformal and duality-invariant non-linear modification of Maxwell's electrodynamics (ModMax) and its BI generalization. In the case of root-TTbar deformed supersymmetric non-linear electrodynamics the authors found that the corresponding operator can be written in a manifestly supersymmetric form with the use of an N=1 superspace supercurrent. The Conclusion of the paper contains a number of interesting and important open issues whose solution will contribute to the further development of this novel subject. The paper is well written and is easy to follow. It certainly deserves publication in SciPost Physics. I only have a couple of remarks that the authors might take into account in the final version of the manuscript.

1) When considering the TTbar deformation of a conformal field theory with the operator (2.20), the authors show that this operator is actually proportional to the trace of the energy-momentum tensor of the deformed theory (see eq. 2.27). In particular, this observation means (as can be directly checked) that the stress tensor operator (2.20) of the BI theory and its ModMax generalization satisfy the condition (2.27). In this respect it would be useful if the authors comment in more detail on the possibility of using the trace-T operator instead of the T^2 operator (2.20) for deforming D=4 conformal field theories, which seems to produce the same result. From this perspective the flow equation (2.30) or (2.31) of the "subtracted" theory is simply related to that of the "unsubtracted" one by shifting trace-T to trace-\tilde T, as in (2.29).

A related remark is that in this case the first and the second term in eq. (3.15) are proportional to each other. Therefore, the operator (2.20) and the trace-T operator seem to be equally "relevant" for preserving the no-birefringence condition.

2) In Sections 5.2 - 5.4, when discussing supersymmetric extensions of the root-TTbar operator and their possible uniqueness the authors used the condition (5.15) which only holds "on-shell", i.e. as a consequence of the equation of motion of the auxiliary field D. Since the flow equation for the TTbar deformed theories involves a deformation operator which is determined off-shell, it would be useful if the authors explain why the use of the condition (5.15) is legitimate for the purposes of their analysis.

Requested changes

As in the report

  • validity: top
  • significance: high
  • originality: high
  • clarity: high
  • formatting: excellent
  • grammar: perfect

Author:  Christian Ferko  on 2023-10-09  [id 4029]

(in reply to Report 1 on 2023-07-24)

We thank the referee for reviewing our paper. We have submitted a revised version of the manuscript which incorporates changes to address the points raised in the referee report.

More specifically, in response to the two points mentioned above,

(1) We agree that the $T^2$ operator and the trace of the stress tensor, with an appropriately chosen coefficient, appear to be "equally relevant" for the purposes of our analysis. In fact, this is true for any one-parameter deformation of a conformal model, which would necessarily satisfy a trace flow equation of the form in our equation (2.19). Thus one could trade any deformation of a conformal seed theory by an operator with a dimensionful coupling constant for a deformation by the trace.

However, the reason that this observation is misleading is that the coefficient of the trace $T^\mu_{\; \mu}$ in this deformation is proportional to $\frac{1}{\lambda}$. Therefore this operator becomes ill-defined in the limit as $\lambda \to 0$, when both the trace vanishes and the coefficient $\frac{1}{\lambda}$ diverges. In this limit, it is not possible to uniquely reconstruct the deforming operator if one presents the flow in terms of the trace. We therefore view the deformation in terms of the $T^2$ operator (or a more general deformation by any $\mathcal{O}$) as more fundamental than the trace, since it can be used to define the leading-order flow around the seed theory.

We have added further comments in the manuscript to clarify this point for the reader. See in particular the discussion around equation (2.20) in the revised draft, as well as the new sentence at the end of the paragraph following equation (3.15).

(2) The referee is indeed correct that the deforming operator is determined off-shell. It would not be legitimate to perform an analysis in which all of the on-shell conditions, including those for the physical fields, are imposed. Many features trivialize when all fields are put on-shell; for instance, the Lagrangian is a total derivative by definition.

However, we underline that our analysis is equivalent to applying on-shell conditions only for non-physical fields, such as the auxiliary field of the $\mathcal{N}=1$ vector multiplet. This point was discussed in other papers of ours, for supersymmetric $T\bar{T}$-like flows in $1\leq D \leq 4$ dimensions. We stress that the auxiliary ${\sf D}$ of the $\mathcal{N}=1$ vector multiplet that we are considering in our paper is not a propagating degree of freedom, and one can freely integrate it in or out without affecting the physics of the model. Said in another way, two flow equations which differ only by the choice of whether to put the auxiliary on-shell will lead to identical deformed equations of motion for the physical, non-auxiliary fields.

To clarify this point, we have extended the discussion in section 5.2, specifically between equations (5.15) and the new (5.19). By looking at the bosonic truncation of the supersymmetric ModMax theory, we have highlighted how it is necessary to integrate out the auxiliary field ${\sf D}$ to obtain the expected root-$T\bar{T}$ flow.

Sincerely,
Christian Ferko, Liam Smith, and Gabriele Tartaglino-Mazzucchelli

---

## Round 2 · Referee Report · Anonymous (Referee 2) · 2023-8-21

Strengths

  1. The paper addresses an interesting and very active field of research, namely the deformations of QFTs constructed out of the stress-energy tensor.
  2. The paper is very well-written, providing an overview of a rapidly-expanding field before going on to present it own results.
  3. The results are novel and interesting.
  4. The paper lays out several interesting questions for further investigation.

Weaknesses

  1. No major weakness

Report

The paper is concerned with the study of deformations of QFTs induced by composite operators built out of the stress-energy tensor. Of these the most famous is the TTbar operator, but several variations on the theme have also been considered more recently --- in particular the "Root-TTbar" operator to which the authors also devote a considerable part of the paper. The authors are chiefly discussing the classical field theory aspects of these deformations in two-dimensions or more. This is not surprising because, with the only exception of the TTbar operator in D=2, all other deformations remain quite poorly understood at the quantum level (mostly due to the intricacies of QFT in D>2).

The paper has two main focuses. The first is to consider some specific and notable example of classical actions which solve the flows under consideration and discuss their properties (in particular birefringence); these are theories which are interesting in their own right, such as Nambu-Goto, Plebanski and Born-Infeld actions. Then the authors look at the compatibility of supersymmetry with these deformations and discuss some of their supersymmetric extensions.

The paper is interesting and well-written. It is certainly deserving of publication. I only have some small comments for the authors.

Firstly, as a minor fixation of mine, I would ask the authors to be more explicit in disclaiming that certain constructions or properties which they review (such as conformality of Root-TTbar deformations) are only currently understood at the classical level. I think that this would be of service to readers unfamiliar with the field.

Secondly and lastly, I would like them to expand on the supersymmetrization of Root-TTbar deformations. The original deformation has the form (following the authors' notations)
$\mathcal{O}=\sqrt{-\text{det}[\hat{T}_{\mu\nu}]}= \sqrt{T_{++}}\sqrt{T_{--}}$
where of course in the last equality I have been intentionally cavalier with the square roots. This is very suggestive to me, and I wonder whether a similarly manifestly factorized form of the deformation may be written in the supersymmetric case (at least for a D=2 SCFT). This might give some insights on the quantisation of the model.

Aside for these minor remarks, I am very happy with the paper and I recommend it for publication.

Requested changes

Better clarify when referring to classical vs. quantum properties; discuss in more detail the supersymmetric extension of Root-TTbar deformations.

  • validity: top
  • significance: high
  • originality: high
  • clarity: top
  • formatting: perfect
  • grammar: perfect

Author:  Christian Ferko  on 2023-10-09  [id 4030]

(in reply to Report 2 on 2023-08-21)

We thank the referee for reading our manuscript and for his or her comments.

In response to the suggested improvements,

(1) We have added additional caveats to warn the reader that our results only hold for classical deformations of the Lagrangian, and not at the quantum level. This includes an additional paragraph just before the beginning of subsection 3.1, which makes a blanket statement that we consider only classical flows, and a more specific clarification just after equation (3.73), which reminds the reader that the root-$T^2$ flow considered in our work only preserves classical conformal invariance.

We agree that these additional clarifications can be useful to help the reader understand the context of the results and highlight that very little is known about the quantum mechanical properties of these deformations, which is an interesting direction for future work.

(2) The referee's point about factorization of the $2d$ root-$T \overline{T}$ operator, and whether any such structure persists in the supersymmetric case, is very interesting. In the current manuscript, we have made only brief comments about the $2d$ analogue of the supersymmetric root-$T \overline{T}$ operator in the conclusion section. However, we have work in progress which will investigate the properties of such $2d$ supersymmetric root-$T \overline{T}$ deformations in more detail. Based on our preliminary results, it appears that the expected factorization does not quite work in the same way when the operator is expressed in terms of supercurrents. We hope to be able to comment on this issue in more detail in future work.

We believe that these modifications have improved the overall quality of the paper and we are grateful to the referee for the suggestions.

Sincerely,
Christian Ferko, Liam Smith, and Gabriele Tartaglino-Mazzucchelli

---

## Round 3 · Referee Report · Anonymous (Referee 1) · 2023-10-9

Report

The authors properly responded to and incorporated into the revised version of their paper the answers to the remarks of my first report

Requested changes

No

---

## Round 3 · Referee Report · Anonymous (Referee 2) · 2023-10-13

Report

The authors have satisfactorily addressed the remarks made in my first report.

---

## Round 3 · Author Response

We are grateful to the two referees for their comments on the first submission of this manuscript. Attached with this message, we are resubmitting a revised paper which incorporates the improvements suggested by the first round of peer review. We believe that these changes have strengthened the paper and we hope that our work will now be suitable for publication in SciPost Physics.

---

## Round 3 · List of Changes

(1) Added a discussion around equation (2.20), and a sentence in the paragraph following equation (3.15), to clarify that the trace flow equation does not uniquely identify the operator driving a flow because it is indeterminate as $\lambda \to 0$, and is thus less fundamental.

(2) Extended the remarks in section 5.2, in particular between equations (5.15) and (5.19), to explain that the on-shell conditions which we use to analyze the root-$T^2$ flow are equivalent to using only the equation of motion for the auxiliary field.

(3) Added comments at the beginning of subsection 3.1, and after equation (3.73), to remind the reader that our results hold only for classical deformations of the Lagrangian.

(4) Clarified that the checks that the $T^2$ deformation preserves the zero birefringence condition, and that any stress tensor deformation preserves duality invariance, were only performed to first order in the deformation parameter.

---

## Editorial Decision

published